# 53BP1 can limit sister-chromatid rupture and rearrangements driven by a distinct ultrafine DNA bridging-breakage process

Ankana Tiwari[1], Owen Addis Jones[1] & Kok-Lung Chan [1]

Chromosome missegregation acts as one of the driving forces for chromosome instability and cancer development. Here, we find that in human cancer cells, HeLa and U2OS, depletion of 53BP1 (p53-binding protein 1) exacerbates chromosome non-disjunction resulting from a new type of sister-chromatid intertwinement, which is distinct from FANCD2-associated ultrafine DNA bridges (UFBs) induced by replication stress. Importantly, the sister DNA intertwinements trigger gross chromosomal rearrangements through a distinct process, named sister-chromatid rupture and bridging. In contrast to conventional anaphase bridge-breakage models, we demonstrate that chromatid axes of the intertwined sister-chromatids rupture prior to the breakage of the DNA bridges. Consequently, the ruptured sister arms remain tethered and cause signature chromosome rearrangements, including whole-arm (Robertsonian-like) translocation/deletion and isochromosome formation. Therefore, our study reveals a hitherto unreported chromatid damage phenomenon mediated by sister DNA intertwinements that may help to explain the development of complex karyotypes in tumour cells.

[1] Chromosome Dynamics and Stability Group, Genome Damage and Stability Centre, University of Sussex, Brighton BN1 9RQ, UK. Correspondence and requests for materials should be addressed to K.-L.C. (email: koklung.chan@sussex.ac.uk)

Gross chromosome rearrangements, as a result of chromosomal instability (CIN) is a hallmark of most, if not all, tumour cells; however, the underlying mechanism is not fully understood. It is generally accepted that CIN contributes to the initiation of tumorigenesis, metastasis progression and multidrug resistance[1,2]. One of the major causes of CIN can be attributed to defects in mitosis such as chromosome misalignments and chromatid non-disjunction, which manifest in the form of lagging chromosomes and anaphase bridges. Generally, lagging chromosomes are generated because of kinetochore-microtubule attachment errors, which not only leads to imbalanced chromosome transmission[3], but also to structural chromosome rearrangements in both a cytokinesis-dependent and cytokinesis-independent manner[4,5]. Additionally, anaphase bridges are generated by abnormal configurations of chromosomes, such as fusions of chromosomes/sister-chromatid arms, or via dysfunctional telomeres[6]. It has been proposed by McClintock that anaphase bridges drive chromosomal rearrangements through a so-called breakage-fusion-bridge (BFB) cycle, where multiple rounds of the joined chromatid bridges break apart during telophase (or cytokinesis) and re-fusing occurs[7,8]. Recently, an elegant study has shown that the breakage of chromatin bridges can be triggered by a cytoplasmic nuclease, TREX1, at telophase-G1 transition and leads to chromothripsis[9].

Previously, we and others have shown that replication of stress-induced DNA entanglements, which are associated with the FANCD2/I dimer, can be carried into mitosis, manifesting as so-called ultrafine DNA bridges (UFBs) in human anaphase cells[10–15]. The resolution of which also leads to DNA damage in the daughter offspring cells[16–18]. It is speculated that this is a result of the separation of DNA intertwining structures at under-replicated regions between sister chromatids[19]. Therefore, the accumulation of DNA entanglements arising during DNA replication and/or homologous recombination (HR) should be limited; otherwise, this could pose substantial threats to chromosome segregation and genome integrity. It is conceivable that this could be more problematic to cancerous cells that bear high intrinsic DNA replication/recombination activities. In fact, a recent study has shown the association of replication stress and CIN[20]. Nevertheless, it remains enigmatic how ultrafine DNA bridging structures may affect faithful chromosome segregation and genome stability.

Here, we have determined that human cancer cells (HeLa and U2OS) rely heavily on a non-homologous end-joining (NHEJ) factor 53BP1[21,22], for chromosome segregation, by limiting the formation of a new type of sister DNA intertwining structure that is not associated with FANCD2, but is dependent of RAD51. Intriguingly, we demonstrate that these sister DNA entanglements drive a novel chromatid damage phenomenon, which induces a rupture of the sister-chromatid axes prior to the breakage of the intertwining DNA bridges. As a result, the ruptured sister chromatids remain tethered by the ultrafine DNA molecules and failed to fully disjoin. Depending on the rupture-bridging positions, this process drives typical and signature chromosome rearrangements, including whole-arm (Robertsonian-like) translocations and isochromosome formation, which are commonly observed in tumour cells. The chromatid rupture-bridging phenomenon is also observed in several unmodified cancer cell lines, suggesting that this alternative mitotic damage action may contribute to the evolution of their karyotypes. In this study, we reveal a new ultrafine DNA bridge-breakage process that drives gross chromosomal rearrangements in cultured human cancer cells, which is regulated by 53BP1.

## Results

### 53BP1 co-localises adjacently to FANCD2 in normal S phase.
The Fanconi anaemia (FA) pathway is activated during S-phase

progression[23]. Previously, we showed that, under replication stress, foci of the FANCD2/I heterodimer persist into mitosis, and subsequently associates with a subclass of UFBs in anaphase cells[10]. Furthermore, the defects in the FA pathway increase chromosome missegregation[11], implying their roles in the formation of DNA intertwining structures. Unresolved DNA entanglements can interfere with faithful chromosome segregation and genome stability. Therefore, to gain insight into how cells prevent DNA entanglements arising during replication, we searched for proteins that co-localise with FANCD2 during unperturbed S phase. We found that 53BP1 forms spontaneous nuclear foci during DNA replication in both normal diploid and cancer cells (Supplementary Fig. 1a, b), where more than half of them surround the FANCD2 foci (Supplementary Fig. 1c, d). This observation suggests that 53BP1 may also participate in the process of DNA replication and/or HR.

### Generation of 53BP1Δ and 53BP1hypo cancer and normal cells.
To explore the role of 53BP1, we generated 53BP1 knockouts in HeLa cervical carcinoma cells, U2OS osteosarcoma cells and hTERT-immortalised RPE1 diploid cells by CRISPR-cas9 genome editing technology. Two guide-RNAs targeting exon 2 and exon 14 of 53BP1 were used. Targeting exon 2 failed to eliminate 53BP1 expression completely in HeLa and RPE1 cells, where residual full-length like protein, and/or small 53BP1 foci were still detectable (Supplementary Fig. 2a–d). However, exon 2 targeting successfully eliminated 53BP1 in U2OS cells (Supplementary Fig. 2e, f). In contrast, targeting exon 14 efficiently eliminated 53BP1 expression in the above three cell lines (Supplementary Fig. 2c, e–h). DNA sequence analysis on the 53BP1 hypomorphic (53BP1hypo) HeLa cells detected no wild-type exon 2 sequence, but three new mutations; all leading to premature translation termination (Supplementary Fig. 3a–c). We thus speculated that the 53BP1 hypomorphic expression in HeLa and RPE1 cells might be due to a leaky expression through a downstream alternative translation site (Supplementary Fig. 3d). Collectively, the 53BP1Δ and 53BP1hypo in both cancer and normal cell lines provide us with useful tools to dissect the functions of 53BP1 during DNA replication.

### 53BP1 depletion in HeLa and U2OS cells compromises chromosome segregation and cell growth.
In the absence of any exogenous DNA assaults, we found that knocking out 53BP1 (53BP1Δ) caused pronounced chromosome missegregation phenotypes, including anaphase bridge and lagging chromatin formation in both U2OS and HeLa cancer cells (Fig. 1a, b). These mitotic defects were also observed in the HeLa cells expressing hypomorphic 53BP1 protein (Fig. 1b). In addition, all 53BP1-depleted U2OS and HeLa cancer cells, including the 53BP1hypo cells, displayed apparent proliferation retardation (Supplementary Fig. 4a, b). Notably, 53BP1hypo HeLa cells grew slightly better than the complete 53BP1Δ cells (Supplementary Fig. 4c), suggesting that a residual activity of 53BP1hypo protein may remain. The mitotic and growth defect phenotypes, however, were either not or only moderately detected in 53BP1hypo and 53BP1Δ RPE1 cells (Fig. 1c & Supplementary Fig. 4d), indicating that HeLa and U2OS cells exhibit a higher reliance on 53BP1 for optimal cell division. Stable overexpression of an EGFP-tagged 53BP1 in 53BP1hypo HeLa cells largely rescued the phenotypes of slow growth and anaphase bridges, but unexpectedly, not lagging chromatin formation (Supplementary Fig. 5 & see explanation in Fig. 5f–h below).

53BP1 has been shown to facilitate DNA double-stranded break (DSB) repair mediated by the NHEJ pathway. We, therefore, investigated whether the missegregation phenotypes

observed might relate to the incompetency of NHEJ. In agreement to other reports[22], we found that 53BP1Δ HeLa (D4 & D10) and U2OS (B4, B18, D29 and D30) displayed increased sensitivities to ionisation radiation (IR) treatments (Supplementary Fig. 6a, b). Similarly, moderately increased IR sensitivities were also detected in RPE1 53BP1Δ cells (D6 and D24) (Supplementary Fig. 6c), although they did not show severe

mitotic and growth defects. Intriguingly, we found that HeLa 53BP1hypo cells, which exhibited elevated chromosome missegregation, were not sensitive to IR treatments (Fig. 1d). As predicted, we observed colocalisation of γH2AX and hypomorphic 53BP1 protein at damage foci, but of a much smaller size (Fig. 1e). In addition, the recruitment of 53BP1 to its binding partner, hRIF1[24–26], was still evident in these cells

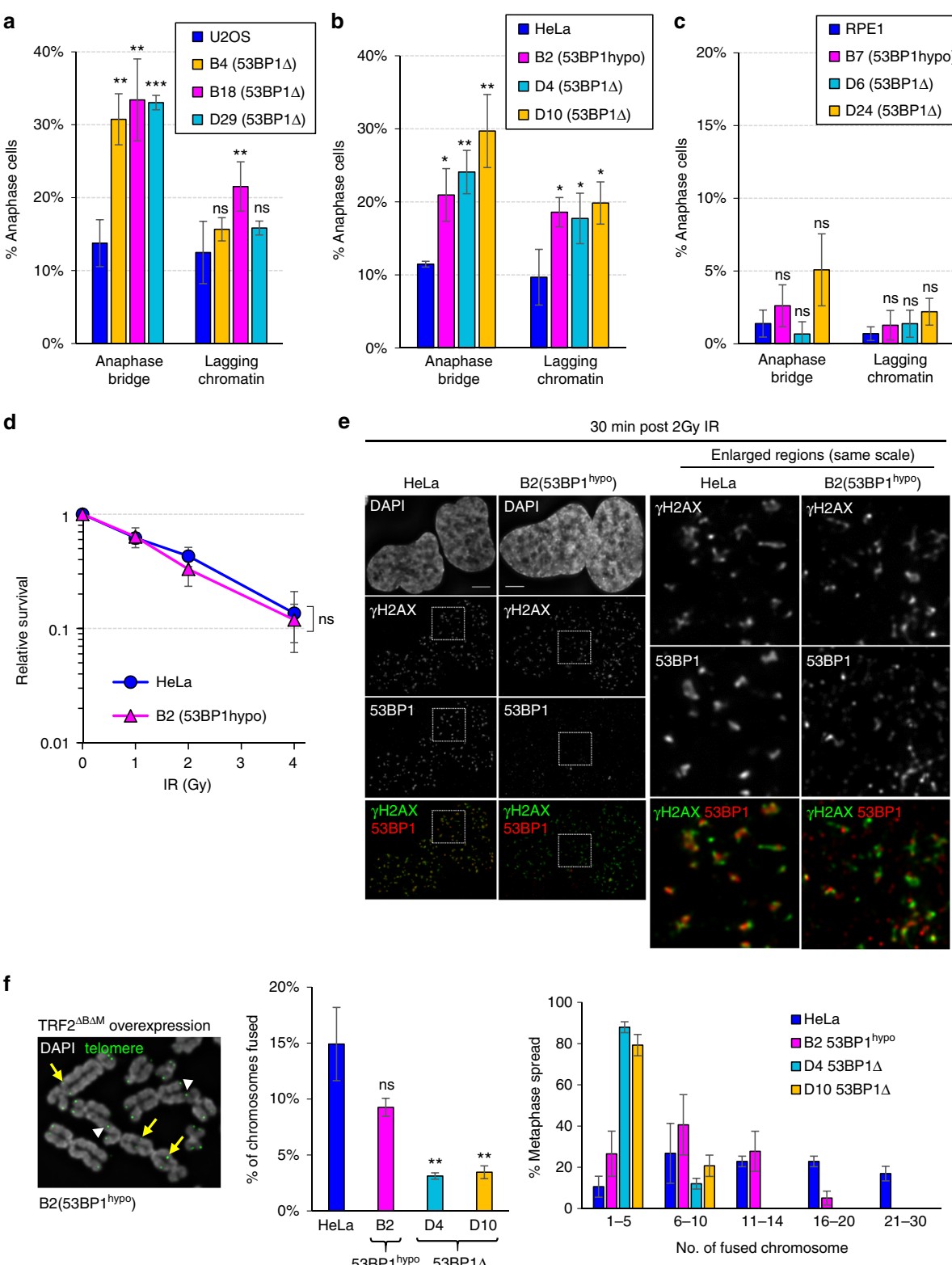

(Supplementary Fig. 6d). Consistently, RPE1 53BP1[hypo] cells also maintained IR resistance despite having very-low 53BP1 expression (Supplementary Fig. 6e). Telomere-end fusion assays revealed that HeLa 53BP1[hypo] cells still exhibited a higher activity of end-joining compared with 53BP1Δ cells, but slightly lower than the parental HeLa cells (Fig. 1f).[27,28] These results indicate that the hypomorphic 53BP1 protein remains competent, at least partially, in NHEJ repair of exogenously induced DSBs. Therefore, we conclude that the increased susceptibilities to the mitotic defects in HeLa and U2OS cancer cells after 53BP1 depletion cannot be merely attributed to NHEJ incompetency. The high reliance of 53BP1 activities in these cancer cell lines, is likely not only because of its necessity for NHEJ repair, but also for a function in facilitating chromosome segregation under physiological growth conditions.

**53BP1 suppresses sister DNA intertwining in cultured cancer cells.** We noticed that there were distinct chromatid non-disjunction features in 53BP1-depleted cancer cells, namely a delay of chromosome separation, manifesting as bridge-like structures (Fig. 2a; arrows) and the formation of multiple lagging chromatin, notably existing as a symmetric pair (Fig. 2a, b; arrowheads). These non-disjunction patterns indicated that they might be caused by ultrafine DNA intertwinements that we and the others previously identified[12,13]. Immunofluorescence staining of UFB-binding proteins, such as PICH and hRIF1[12,29], revealed that the characteristic anaphase bridge(-like) structures and lagging chromatin pairs were indeed tethered by UFBs (Fig. 2c, d). As predicted, the frequency and number of UFBs were significantly increased in both 53BP1Δ and 53BP1[hypo] cancer cells (Fig. 2e, f), suggesting that 53BP1 is required for the suppression of UFB formation.

Chromatin bridges in anaphase can be caused by inter-chromosomal linkage formed in dicentric or radial chromosomes, or via sister-chromatid intertwining. To distinguish these, we developed a protocol using EdU to differentially label one of the two sister chromatids in mitotic cells (Fig. 3a, b). To avoid complications arising from NHEJ malfunction, we performed most of our investigation using HeLa 53BP1[hypo] cells, which, as demonstrated above, largely maintain the NHEJ pathway. Remarkably, almost all anaphase bridges (98%) in the HeLa 53BP1[hypo] cells displayed a symmetric (but opposite) staining pattern (Fig. 3c), showing either EdU labelling on one-half of the DNA bridge (Fig. 3d), or resembling sister-chromatid exchange (SCE) patterns (Fig. 3e). Similarly, the lagging chromatin pairs also displayed the symmetric labelling patterns (Fig. 3f), highly suggesting that the non-disjoined chromatin structures are composed of sister chromatids. Therefore, we conclude that 53BP1 acts to suppress DNA intertwinements, arising mainly between sister chromatids. In agreement with this conclusion, dicentric or radial chromosomes were rarely observed in

metaphase spreads of both 53BP1[hypo] and 53BP1Δ HeLa cells (Fig. 3g). We henceforth name this phenomenon as "sister-chromatid bridging" to distinguish it from the general terminology of anaphase bridges.

**The FANCD2 non-associated sister UFB is RAD51-dependent.** DNA catenation, (sister-)telomere fusion or replication stress can lead to UFB formation[19,30–32]. The fact that 53BP1 accumulates adjacently to FANCD2 foci during S phase led us to speculate that its deficiency might exacerbate replication stress particularly in these cancer cells, mimicking the effect of DNA polymerase inhibition induced by aphidicolin treatments, which causes the accumulation of late replication intermediates (LRIs) and the formation of UFBs positive for FANCD2 foci[10,11] (Supplementary Fig. 7a). Contrary to our hypothesis, we found that 53BP1 depletion in HeLa and U2OS cells significantly increased anaphase populations having FANCD2-negative UFBs (Fig. 4a). In contrast to aphidicolin-induced replication stress, it mainly increased anaphase cells with FANCD2-positive, but not the FANCD2-negative UFBs (Supplementary Fig. 7a). Consistently, most of the UFBs (56–86%) detected in the 53BP1-depleted cells are FANCD2 negative, except when the 53BP1[hypo] cells were pre-treated with aphidicolin, which increased the proportion of UFBs positive for FANCD2 foci from 14 to 55% (Fig. 4b & Supplementary Fig. 7b). These results suggest that a new subclass of sister DNA bridge arises in these cancer cells when 53BP1 activities become limiting. Moreover, this also indicates that the FA pathway is not compromised in 53BP1[hypo] cells, which is further supported by the fact that, like the parental HeLa, aphidicolin treatment readily elevated mitotic FANCD2 foci in the 53BP1[hypo] cells (Supplementary Fig. 7c, d). The lack of increased replication stress phenotypes, including spontaneously elevated FANCD2 mitotic foci, abnormal S phase accumulation and increased common fragile site (CFS) expression (Supplementary Fig. 7c–f), highly suggests that the induction of FANCD2-negative UFB formation in the 53BP1-depleted cells is unlikely caused by the same mechanism of replication stress, induced by DNA polymerase inhibition.

A very characteristic feature of the anaphase bridge observed in the 53BP1[hypo] cells is that the sister-chromatid arms are tethered by UFBs. We found that nearly 75% of the associated chromosomes were positive for γH2AX signal (Fig. 4d, e), implicating a DNA damage response acting in this phenomenon. The fact that most of the UFB-tethered chromatin bridges arise originally from sister chromatids and negative of FANCD2 binding infers that their formation may associate with HR activity. To test this, we knocked down RAD51, a key initiation factor of HR[33] in HeLa 53BP1[hypo] cells (Fig. 4f). As shown previously, RAD51 depletion led to increased replication stress[10], and hence elevated the anaphase population having FANCD2-postive UFBs (Fig. 4g, h). Crucially, the RAD51 knockdown

---

**Fig. 1** 53BP1 depletion leads to increased chromosome non-disjunction in human cancer cells. Quantitation of anaphase bridge and lagging chromatin in 53BP1Δ and 53BP1[hypo] cells **a** U2OS, **b** HeLa and **c** RPE1. Numbers of cell counted: U2OS = 529, B4 = 327, B18 = 344, D29 = 372; HeLa = 540, B2 = 351, D4 = 426, D10 = 317; RPE1 = 435, B7 = 449, D6 = 382, D24 = 458 from 3–4 separate preparations. **d** IR sensitivity assay on HeLa and B2 (53BP1[hypo]) cells (N = three independent experiments). Statistical significance was determined by two-way ANOVA. **e** The formation of IR-induced DNA damage foci in HeLa and B2 (53BP1[hypo]) cells. Thirty minutes post 2 Gy IR, the cells were immunostained with anti-53BP1 and anti-γH2AX antibodies. Enlarged regions demonstrating the recruitment of 53BP1 (red) at the IR-induced DNA breaks, marked by γH2AX (green). **f** Representative image of telomere fusions on metaphase chromosomes of B2 (53BP1[hypo]) cells overexpressing TRF2[ΔBΔM]. An example of fusions on single (arrowheads) and both (arrows) sister telomeres indicated (left). Middle: percentage of chromosome fusion events in HeLa, B2 (53BP1[hypo]), D4 and D10 (53BP1Δ) cells, >75 metaphases of each cell line were analysed from three independent experiments. Right: histogram showing telomere fusion events in HeLa, B2 (53BP1[hypo]), D4 and D10 (53BP1Δ). Total number of chromosomes analysed in HeLa = 3890, B2 = 3792, D4 = 4720 and D10 = 4790 from >60 metaphase spreads. Statistical significance was determined by T-test (* p < 0.0, ** p < 0.01, *** p < 0.001, ns nonsignificant). Error bars represent s.d. of three independent experiments. Scale bars, 5 μm

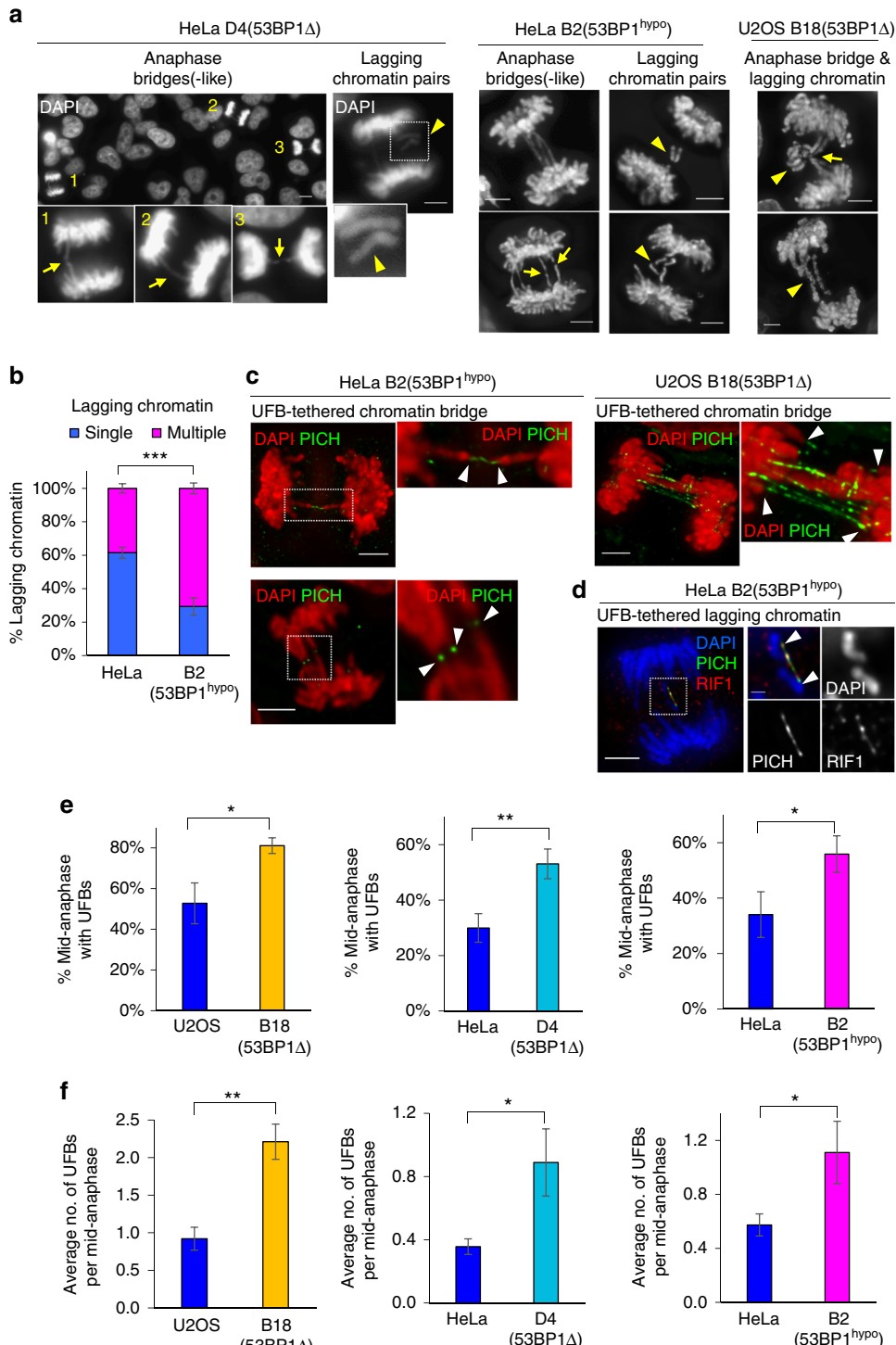

**Fig. 2** Chromosome non-disjunction in 53BP1-depleted cancer cells is mediated by ultrafine DNA bridges. **a** Representative images of DAPI-stained HeLa D4 (53BP1Δ), HeLa B2 (53BP1$^{hypo}$) and U2OS B18 (53BP1Δ) cells showing anaphase bridges, bridge-like structures and lagging chromatin pairs (arrowheads). Insets show enlarged view of the numbered cells exhibiting bridge-like (arrows; 1 & 3), bridge structures (arrow; 2). **b** Increased formation of multi-lagging chromosomes in HeLa B2 (53BP1$^{hypo}$) cells as compared to HeLa. Quantification of single or multiple lagging chromatin in HeLa and B2 (53BP1$^{hypo}$) cells. more than 100 cells with lagging chromatin were counted from three separate preparations. **c** Deconvolved high-resolution images showing the two separating chromatin arms (red) connected by PICH-UFBs (green) in HeLa B2 (53BP1$^{hypo}$, Left) and in U2OS B18 (53BP1Δ, Right). Insets shows enlarged views of the selected region. **d** Deconvolved image showing hRIF1 (red) localises at a PICH-coated UFB (green), intertwining a pair of lagging chromatin in HeLa B2 (53BP1$^{hypo}$). Inset shows enlarged view of selected region. **e** Percentage of mid-anaphase cells with PICH-UFBs in U2OS B18 (53BP1Δ), HeLa D4 (53BP1Δ) and HeLa B2 (53BP1$^{hypo}$). Numbers of anaphase counted: U2OS = 91, B18 = 90; HeLa = 123, D4 = 105; HeLa = 138, B2 = 139 from three separate preparations. **f** Average number of PICH-UFB per mid-anaphase cell in U2OS B18 (53BP1Δ), HeLa D4 (53BP1Δ) and HeLa B2 (53BP1$^{hypo}$). Error bars represent s.d. of three independent experiments. Statistical significance was determined by T-test (* $p < 0.05$, ** $p < 0.01$, *** $p < 0.001$). Scale bars, 5 μm

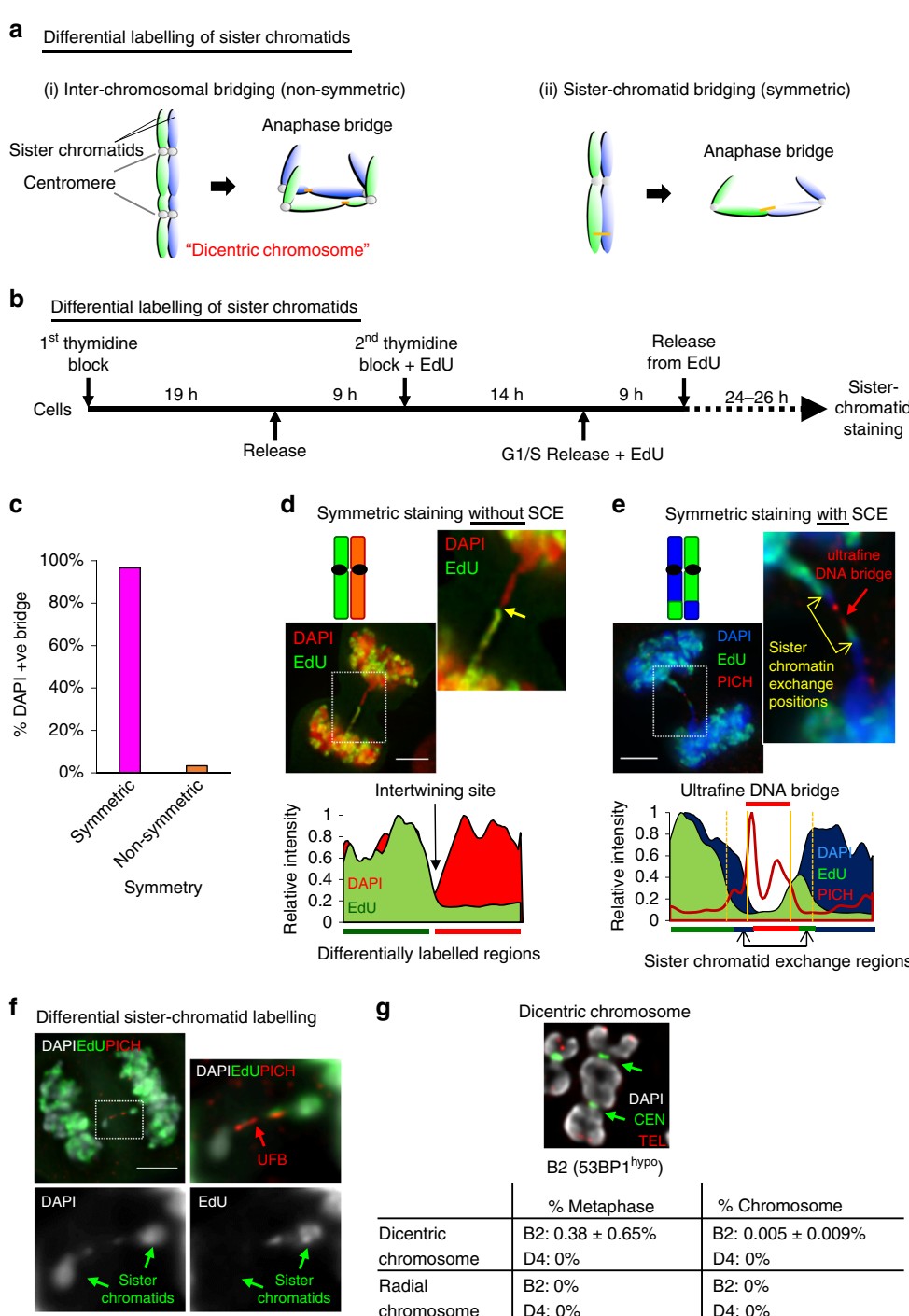

**Fig. 3** DNA entanglements between sister chromatids are the major cause of chromosome non-disjunction in 53BP1[hypo] cells. **a** Diagram depicts differential sister-chromatid labelling (green; EdU) to distinguish between anaphase bridges caused by (i) inter-chromosomal fusion, shown as non-symmetric or by (ii) DNA intertwining between sister chromatids, shown as symmetric. **b** Experimental setup of differentially labelling sister chromatids. **c** Quantitation of anaphase DAPI bridges with symmetrical and non-symmetrical staining patterns in HeLa B2 (53BP1[hypo]) cells. Over 65 anaphase cells with chromatin bridges were counted from two independent experiments. Total numbers of DNA bridge (n = 74) were counted. **d** Representative image of a HeLa B2 (53BP1[hypo]) cell showing a symmetrical (but opposite) staining pattern along a DAPI anaphase bridge (Top). Yellow arrow in the inset shows EdU (green) staining was present on only one-half of the bridge. Area chart displaying the relative intensity along the differentially labelled sister chromatids (Bottom). **e** Representative example of a chromatin bridge, linked by a PICH-UFB (red), displaying a staining pattern resembling sister-chromatid exchange (SCE) (Top). Yellow arrows in the inset indicates the sister-chromatid exchange positions. Area chart displaying the relative intensity of differentially labelled sister chromatids with sister-chromatid exchange regions (Bottom). **f** A pair of lagging chromosomes showing differential staining (green arrow) were inter-linked by a PICH-UFB (red arrow) in HeLa B2 (53BP1[hypo]) cells. **g** Quantitation of dicentric and radial chromosomes in HeLa B2 (53BP1[hypo]) and HeLa D4 (53BP1Δ) metaphase cells. Top: an example of a dicentric chromosome in B2 (53BP1[hypo]) cells (arrows indicate centromeres). Over 6000 chromosomes in 88 metaphase spreads from three independent experiments were examined. Chromosomes were hybridised with telomere and centromere PNA FISH probes. DNA was stained with DAPI. Scale bars, 5 μm

significantly diminished the percentage of the 53BP1hypo anaphase cells having the FANCD2-negative UFBs (Fig. 4i). These results indicate that the increased formation of FANCD2-negative sister DNA bridges in 53BP1hypo cells is dependent on the HR activity. It is plausible that (partial) loss of 53BP1 function may increase a distinct type of replication difficulty, which is converted into the sister DNA intertwinements by HR reaction. Alternatively, 53BP1 may prevent the formation of, or facilitate resolution of, HR intermediates (Fig. 4j). In fact, we detected increases in SCEs in the 53BP1hypo HeLa cells (Supplementary

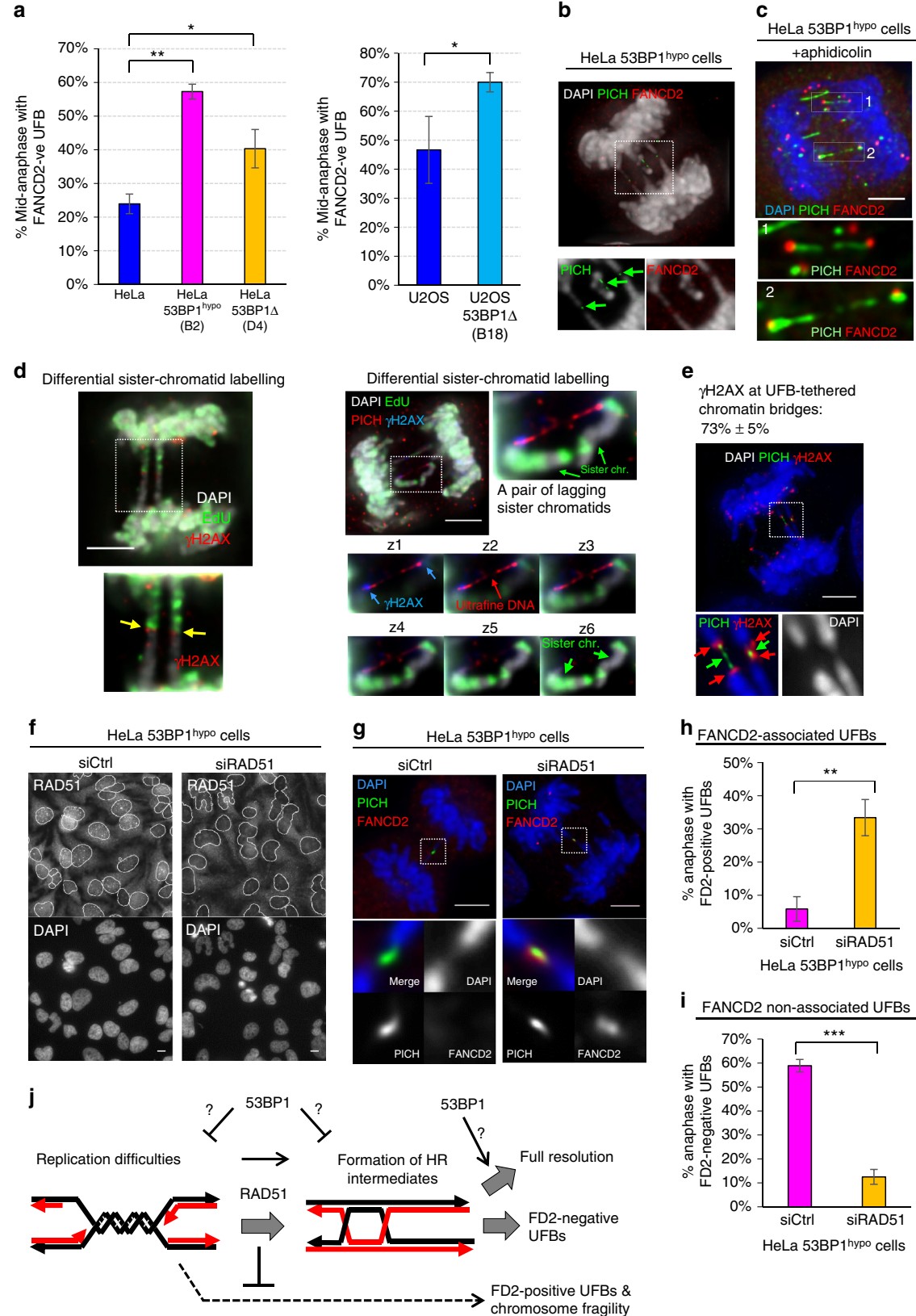

Fig. 8a). However, we cannot rule out that this may be also be due to impairments of non-crossing over resolution activity.

Contrary to our speculation, a recent report suggested that 53BP1 is required for high fidelity of HR repair on double-ended DSBs, probably through the prevention of exacerbated DNA end resection that channels excessive single-stranded annealing (SSA) reaction[34]. However, we did not detect significant changes in the ssDNA formation (as measured by chromatin-bound RPA) in unperturbed 53BP1[hypo] S-phase cells (Supplementary Fig. 8b). It is possible that 53BP1 may influence HR fidelity differently on double-ended and single-ended DSBs, where the latter (associated with replication forks) is not an ideal substrate for SSA even if excessive DNA end resection occurs. On the other hand, loss of 53BP1, as shown previously to rescue HR in *brca1*−/− cells[35], may further relieve the constraints of HR at damaged forks. Nevertheless, our data demonstrate that the HR pathway participates to a certain extent on the formation/accumulation of the FANCD2-negative sister-chromatid bridges in the 53BP1[hypo] cells.

**DNA intertwinements lead to sister-chromatid rupture**. We next characterised this subclass of sister-chromatid bridges and their effects on chromosome integrity. A very striking feature of the FANCD2-negative UFBs is that they were frequently found to emerge at the terminal regions of the separating or lagging sister chromatids, which could represent telomeres[32]. Alternatively, this may be explained as a result of two sister arms being tethered by UFBs, leading to their termini pointing towards each other during segregation (Fig. 5a). According to these hypotheses, telomeres are expected to be always present on anaphase/DNA bridges as shown as an example in Fig. 5b. Surprisingly, telomeres were rarely found on the DNA bridges and, indeed, they were missing at the UFB-tethered sister chromatids (Fig. 5c, d). In contrast, inter-chromosomal fusion generated by overexpression of TRF2[ΔBΔM], as expected, led to the majority of anaphase bridges linking via their telomeres (Fig. 5d, e; arrows), validating our ability to detect such events when they arise. In parallel, we observed the loss of telomeric regions on the lagging chromatid pairs, specifically at the chromosomal termini where the UFBs emerged (Fig. 5f; asterisks). A simple interpretation of these results is that the UFB-tethered sister chromatids (whether they exist as anaphase bridges or lagging chromatin) are broken chromosomes. Most notably, the breakage of the sister-chromatid axes occurred at the sites where the ultrafine DNA linkage emerged and persisted.

53BP1 localises to kinetochores in early mitosis[36]. A previous study has reported that 53BP1 knockdown can cause kinetochore-microtubule attachment errors and the subsequent formation of lagging chromosomes[37]. However, our extensive chromosome analyses in anaphase cells have led us to reveal an alternative explanation for the missegregation phenotype. The facts that genuine intact lagging chromosomes were not detected and the presence of UFB stretching between the lagging chromatin pairs highly indicate that the missegregation is caused by persistent DNA intertwinements rather than kinetochore-microtubule mis-attachment[38]. Because of the unexpected finding of chromatid breakage, we re-examined the failure of an EGFP-53BP1 wild-type protein to suppress lagging chromatin in HeLa 53BP1[hypo] cells (as shown in Supplementary Fig. 5d). We found that the ectopic overexpression of 53BP1 successfully reduced the formation of the broken lagging chromatin pairs (Fig. 5g; single telomere end). However, it also generated extra intact lagging chromosomes (Fig. 5g, h). Thus, the overall anaphase population with lagging chromatin remained unchanged. These data indicate that overexpression, but not depletion, of 53BP1 probably interferes with proper kinetochore-microtubule attachment.

More importantly, we found that the sister DNA bridges not only caused distinct chromatid non-disjunction, but also led to the identification of a new mitotic damage phenomenon, we termed sister-chromatid rupture and bridging. Contrary to conventional anaphase bridge-breakage models, our data clearly show that the occurrence of DNA damage on the intertwined sister chromatids is not coupled to the breakage of the DNA bridges and is independent of cytokinesis.

**Sister-chromatid rupture occurs strictly upon anaphase onset**. One plausible explanation for the appearance of the ruptured (but remaining intertwined) sister chromatids during anaphase is that they have already broken during DNA replication. If this is correct, we should expect to see chromosomes with broken arms or sister-chromatid arm fusion[7] in (pro)metaphase cells (Supplementary Fig. 9a). Of the thousands of metaphase chromosomes analysed on both 53BP1Δ and 53BP1[hypo] HeLa cells, we found that almost all (>99.9%) were in a normal intact configuration (both termini were present). Only <3% of metaphases showed evidence of a chromosome with a sister-arm fusion (Supplementary Fig. 9b), which cannot explain the high fraction (>30%) of anaphase cells harbouring ruptured sister-chromatid bridges or lagging chromatin. Thus, the chromatid breakage is inferred to occur after anaphase onset. Next, we tested if the microtubule pulling on the intertwined chromatids causes the rupture. We triggered premature sister-chromatid separation by knocking down Sgo1 in 53BP1[hypo] (pro)metaphase cells.

**Fig. 4** The formation of sister DNA entanglements in 53BP1-depleted HeLa cells is dependent on RAD51. **a** Quantitation of 53BP1-depleted HeLa (left) and U2OS (right) anaphase cells forming FANCD2-negative UFBs. Numbers of anaphase counted: HeLa = 135, B2 = 112, D4 = 105; U2OS = 91, B18 = 90 from three independent experiments. **b** Maximum z-projection high-resolution image showing multiple short FANCD2-negative PICH-coated UFBs (arrows), linking the separating chromatin and lagging chromosomes in B2 (53BP1[hypo]) cells. Inset shows that PICH stained UFBs (green) are not associated with FANCD2 foci (red). **c** Maximum z-projection high-resolution image showing the association of FANCD2 foci (red) on PICH-UFBs (green) in aphidicolin-treated HeLa B2 (53BP1[hypo]) cells. Inset shows enlarged view of PICH-coated UFBs are positive of FANCD2 foci at their termini. **d** Representative images showing γH2AX present on chromatin bridges and lagging chromatin pairs in HeLa 53BP1[hypo] cells. Left: maximum z-projection image showing γH2AX (red) at the junction (arrows) of the differentially labelled sister-chromatid bridges (EdU; green). Right: a pair of lagging sister chromatin intertwined by a PICH-UFB (red) and positive of γH2AX (blue) at their termini. Bottom Right: panels showing single-plane images of the intertwining lagging sister chromatin. Blue arrows indicate γH2AX present at the tips of the chromatin. **e** HeLa B2 (53BP1[hypo]) cell showing the presence of γH2AX signals (red) at the termini of chromatin that were tethered by PICH-coated UFBs (green). **f** HeLa B2 (53BP1[hypo]) cells were transfected with control or Rad51 siRNA oligos, followed by IF analysis using anti-Rad51. Nuclei are outlined (grey). **g** RAD51 knockdown caused the formation of FANCD2-assoicated (red) PICH-UFBs (green) in HeLa B2 (53BP1[hypo]) cells. **h** Quantitation of HeLa B2 (53BP1[hypo]) anaphase cells with FANCD2-positive UFBs following RAD51 knockdown. **i** Quantitation of HeLa B2 (53BP1[hypo]) anaphase cells with FANCD2-negative UFBs following RAD51 knockdown. Numbers of anaphase cells scored: B2 + control siRNA = 350, B2 + RAD51 siRNA = 220 from three independent experiments. **j** A model showing the potential roles of 53BP1 and RAD51 in the formation of FANCD2-negative sister DNA bridges in the 53BP1-depleted cells. Error bars represent s.d of three independent experiments. Statistical significance was determined by *T*-test (* $p < 0.05$, ** $p < 0.01$, *** $p < 0.001$). Scale bars, 5 μm

Although we detected increase in chromatin breakage on single chromatids (Supplementary Fig. 9c; 10% in HeLa and 15% in B2 53BP1$^{hypo}$ cells), the frequency was still lower than the observed one in anaphase (>30%). Collectively, these data strongly indicate that the rupture of the sister-chromatid axes occurs strictly after anaphase onset, which may infer their damage is mediated by factors requiring APC/C activation rather than merely spindle pulling.

**Sister-chromatid rupture exacerbates chromosomal rearrangements.** A prediction arising from the above findings is that it will lead to chromosomal damage and CIN in the offspring cells.

In agreement with this, we observed increased numbers of G1 53BP1 nuclear bodies in HeLa 53BP1$^{hypo}$ daughter cells (Supplementary Fig. 9d). Additionally, we observed an increase in numerical chromosome alterations. Interestingly, chromosome loss, over gain, seemed to be dominant in all 53BP1$^{hypo}$ clones (Supplementary Fig. 10a). Next, we examined the structural chromosome alterations. Although, the HeLa cancer genome is considered unstable, we found that their karyotypes are relatively stable, as reported previously[39]. Whole chromosome painting and fluorescence in situ hybridisation (FISH) analyses revealed that all HeLa cells maintained four copies of chromosome7 plus one derivative, whereas the majority of them (94%) maintaining three

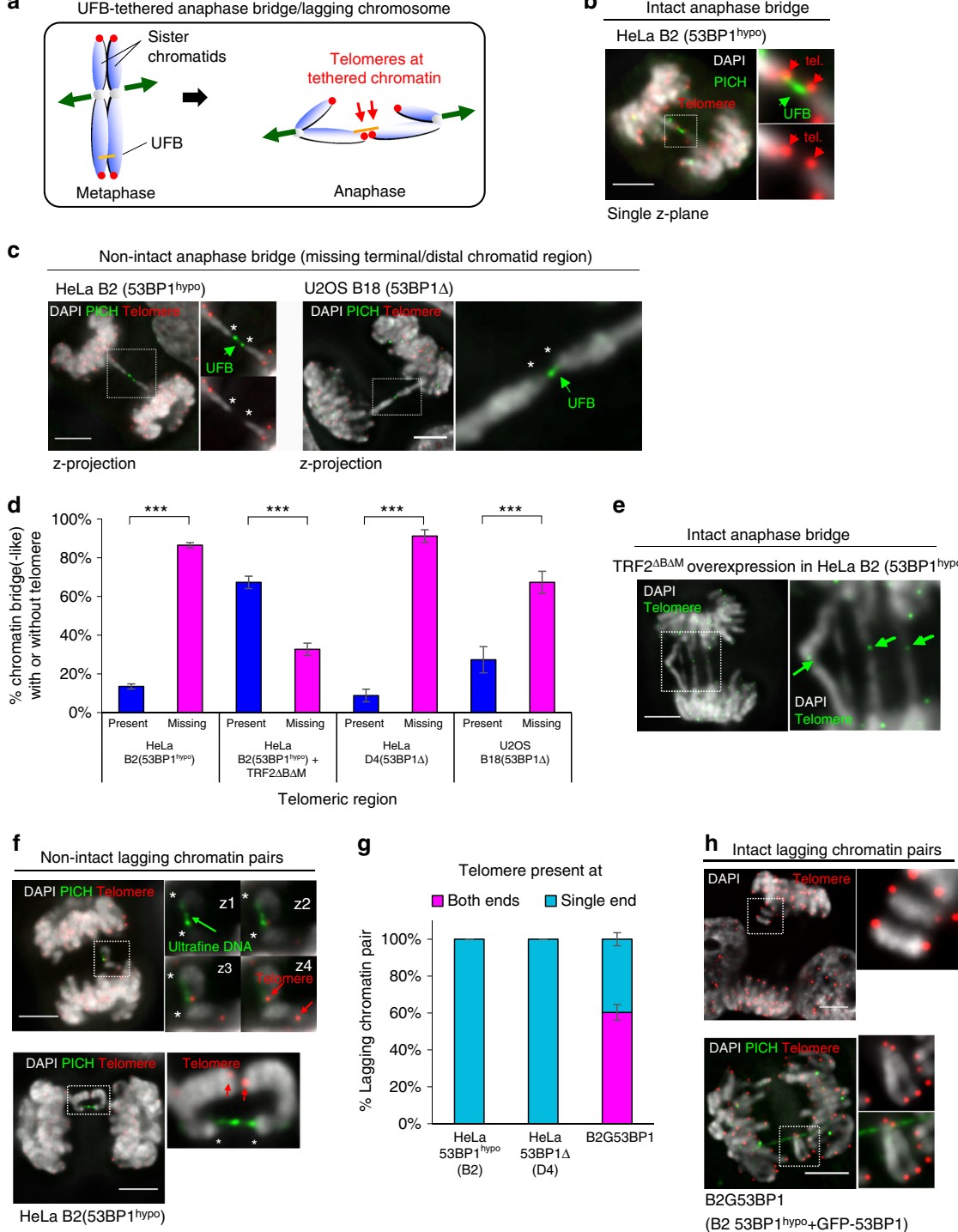

**a** UFB-tethered anaphase bridge/lagging chromosome

**b** Intact anaphase bridge
HeLa B2 (53BP1$^{hypo}$)
Single z-plane

**c** Non-intact anaphase bridge (missing terminal/distal chromatid region)
HeLa B2 (53BP1$^{hypo}$) U2OS B18 (53BP1Δ)
z-projection z-projection

**d** Telomeric region

**e** Intact anaphase bridge
TRF2$^{ΔBΔM}$ overexpression in HeLa B2 (53BP1$^{hypo}$)

**f** Non-intact lagging chromatin pairs
HeLa B2(53BP1$^{hypo}$)

**g** Telomere present at
■ Both ends ■ Single end

**h** Intact lagging chromatin pairs
B2G53BP1
(B2 53BP1$^{hypo}$+GFP-53BP1)

copies of chromosome16 and a 16p derivative, (6% with two chr16 + one derivative) (Supplementary Fig. 10b, c). However, in the 53BP1-depleted (53BP1$^{hypo}$ and 53BP1Δ) HeLa cells, we found a variety of structural rearrangements on these chromosomes. They included deletions and translocations on the distal arm of 16q, chr7 and 16 whole-arm deletion, chr7 and 16 whole-arm (Robertsonian-like) translocations and 16p/16q isochromosome formation (Fig. 6a–e). Crucially, such rearrangements were never observed in the parental HeLa cells (Fig. 6f). A common breakpoint was assigned at a site just upstream of a common fragile site (CFS), FRA16D, in the WWOX locus, while interestingly another one was mapped to, or very close to, centromeres. Together, these results suggest that the sister-chromatid rupture in anaphase can lead to gross chromosome rearrangements.

**Chromosome rearrangements link to sister-chromatid bridging**. To provide further evidence that the observed rearrangements are directly related to the sister-chromatid rupture/bridging, we carefully examined the existence of DNA entanglements at these regions. By extending our FISH probes along the FRA16D/WWOX locus (Fig. 7a), we revealed a hitherto unidentified DNA thread structure linking the sister chromatids in 53BP1$^{hypo}$ HeLa metaphase spreads (but not in the parental HeLa cells), which presumably is a precursor of the sister DNA bridging structures in anaphase. It was mapped at the promoter (91O9), or the gene body (264L1), of the WWOX locus (Fig. 7b–d; arrows and Supplementary Fig. 10d). To our knowledge, this is the first time that a genuine DNA intertwining molecule on cohesed sister chromatids has been visualised and mapped. Importantly, CFS fragility was not observed in the intertwined FRA16D locus, which supports our above data that the intertwining molecule is unlikely originating from an incomplete replication intermediate. By measuring inter-sister locus distances, we determined that the promoter region (91O9) of one WWOX allele displayed the highest frequency of DNA thread formation, something that was absent in a region 2.7 Mb upstream (352J17) (Fig. 7d). Correlatively, this new form of DNA thread structure was found in 53BP1$^{hypo}$ clones (B2 and b9), each of which also harboured the corresponding rearrangements at 16q distal regions in the new derivative chromosome. By contrast, the 53BP1$^{hypo}$ clone (b15) did not show either of these features. These observations demonstrate a high correlation between region-specific rearrangements and sister DNA bridging.

We next investigated whether centromeres, another hotspot of rearrangements, also have high incidence of rupture/bridging. Because centromeres remain cohesed in early mitosis, we examined the chromatin bridges and lagging chromatin structures in anaphase cells. We did not observe centromeres on the UFB-tethered chromatin bridges. Instead, centromeres were detected at the point at which the vast majority of lagging chromatin was intertwined (Fig. 7e, f). Importantly, the centromeres were located at the chromatid ends where the telomeric regions (or the distal arms) were missing, but remained tethered (Fig. 7g). Live-cell imaging of 53BP1$^{hypo}$ (B2) cells stably expressing mCherry-H2B revealed that the lagging chromatin pairs failed to disjoin and co-segregated into the same daughter cell (Supplementary Fig. 10e). This is presumably due to the persistent DNA tethering at their sister centromeres counteracting the spindle-separation force. The co-segregation of ruptured sister whole-arms, thus provides an ideal precursor for isochromosome formation that was found in these cells. Taken together, our study reveals that the illegitimate sister DNA entanglements can drive gross chromosomal rearrangements via a distinct sister-chromatid rupture-bridging action that has never been reported before.

**Sister-chromatid rupture occurs in cultured cancer cells**. We next addressed if this phenomenon is limited in the 53BP1-depleted cancer cells, or occurs generally in other unmodified human cancer cells. We carefully characterised the missegregating chromatin in anaphases of a variety of human cancer cell lines, including HeLa, U2OS, Saos-2 and HCT116, in the absence of exogenous perturbation. As expected, we observed spontaneous chromosome missegregeation in these cells, but most importantly, we detected a subset of non-disjunction following a similar pattern of sister-chromatid rupture and bridging (Fig. 7h; HeLa (10%), U2OS (7%), Saos-2 (8%) and HCT116 (13%). Thus, our results highly suggest that in addition to the conventional anaphase bridge-breakage mechanism, sister DNA intertwining structures may contribute to influence the complex karyotypes during cancer evolution. Indeed, we have determined that the HeLa genome harbours several Robertsonian(-like) translocations or deletions, they involved chromosomes 1, 3, 7, 9, 15 and 16 (Supplementary Fig. 11), and more reported previously[39]. Thus, our study of sister-chromatid rupture and bridging may provide an alternative explanation for their formation.

In summary, we have revealed a distinct sister-chromatid rupture and bridging phenomenon for how ultrafine sister DNA entanglements may drive excessive gross chromosome rearrangements that can be suppressed by 53BP1 in cultured cancer cells (Fig. 8). Our study may provide an alternative explanation of how complex karyotypes arise during cancer developments in the context of illegitimate formation and resolution of sister DNA intertwinements.

**Fig. 5** Sister-chromatid rupture is associated with HR-mediated DNA intertwining. **a** Diagram depicting the formation of telomere-positive DNA bridges, resulting from DNA entanglements between sister chromatids. **b** A single-plane high-resolution image showing the presence of telomeres (red) at the termini of chromatin tethered by an UFB (PICH; green) in a HeLa B2 (53BP1$^{hypo}$) anaphase cell. **c** Maximum z-projection image of an UFB-tethered chromatin bridge missing telomeric regions (asterisks) at their terminal ends in HeLa B2 (53BP1$^{hypo}$) (left) and in U2OS B18 (53BP1Δ) cells (right). Insets show enlarged images of the DNA/chromatin bridges. **d** Quantitation of DAPI bridges with and without telomeres in HeLa, HeLa 53BP1$^{hypo}$, U2OS 53BP1Δ and in HeLa 53BP1$^{hypo}$ cells overexpressing TRF2$^{ΔBΔM}$. Note: Majority of DNA bridges are negative for telomere signals in 53BP1-depleted cells, except after TRF2$^{ΔBΔM}$ overexpression. Total numbers of DAPI bridge analysed were B2 = 85, B2 + TRF2$^{ΔBΔM}$ = 60, D4 = 45 and B18 = 55 from three independent experiments. Statistical significance was determined by T-test (***, $p < 0.001$). **e** A representative image showing telomeres were detected on chromatin bridges induced by telomere end-joining. Inset indicates the presence of telomere signals (green) at chromatin bridges (arrows). **f** Representative images of HeLa B2 (53BP1$^{hypo}$) cells showing UFBs tethering a pair of lagging chromatin at their termini, at where telomere (red) signals are absent (asterisks), but remained connected by PICH-UFBs. Insets showing consecutive single z-plane images of the lagging chromatin. **g** Quantitation of telomeres present at one or both ends of lagging chromatin pairs. Note that all lagging chromatin pairs lack one telomere end in HeLa B2 (53BP1$^{hypo}$) and D4 (53BP1Δ) cells. In contrast, HeLa 53BP1$^{hypo}$ cells stably overexpressing GFP-tagged 53BP1 (B2G53BP1) cells contain lagging chromatin having telomere signals at both, or single ends. Total numbers of lagging chromatin pair analysed were B2 = 35, D4 = 41 and B2G53BP1 = 44, from three independent experiments. **h** Representative images of B2G53BP1 cells showing intact lagging chromosome with telomere signals (red) at both termini. Insets showing enlarged view of the lagging chromatin with telomere signals present at both their termini. Error bars represent s.d of three independent experiments. Scale bars, 5 μm

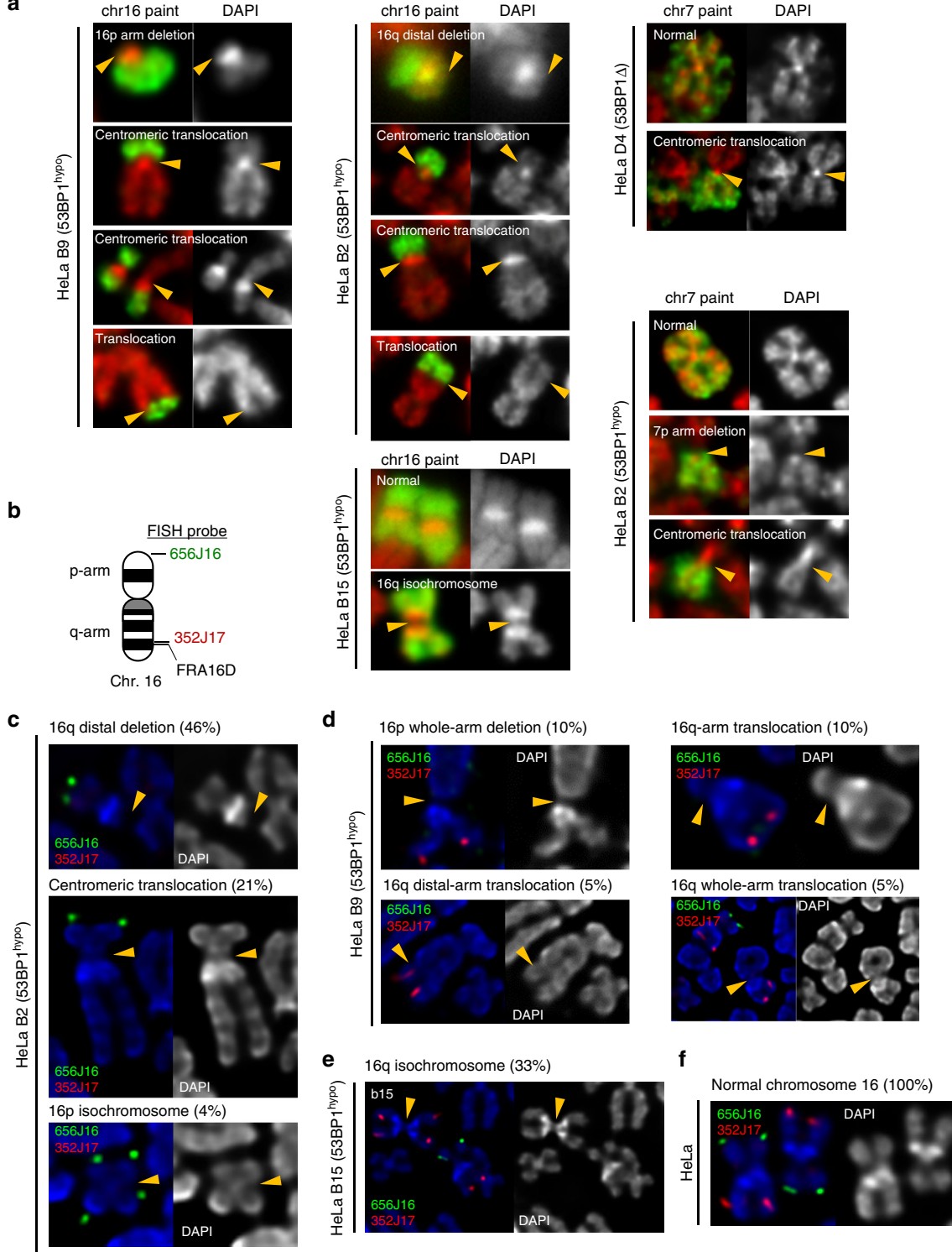

**Fig. 6** Gross chromosomal hyper-rearrangements mediated by sister-chromatid bridging in 53BP1-depleted cancer cells. **a** Formation of new chromosome 16 and 7 derivatives in HeLa 53BP1[hypo] and 53BP1Δ clones. Left panels: whole chromosome 16 painting revealing 16p arm deletion and arm/centromeric translocations in HeLa b9 (53BP1[hypo]) cells. Middle panels: 16q deletions and arm/centromeric translocations in HeLa B2 (53BP1[hypo]) cells; Isochromosome 16q formation in HeLa b15 (53BP1[hypo]) cells. Right panels: whole chromosome 7 painting revealing centromeric translocation in HeLa D4 (53BP1Δ) cells and arm deletion and centromeric translocation in HeLa B2 (53BP1[hypo]) cells. **b** Ideogram of human chromosome 16, marking the positions of FISH probes used in Fig. 6c–f. **c** 16q distal arm deletion, centromeric translocation and 16p isochromosome formation was identified in HeLa B2 (53BP1[hypo]) populations with the indicated percentages. **d** 16p whole-arm deletion, 16q whole-arm translocation and 16q distal-arm translocation were detected in HeLa b9 (53BP1[hypo]) populations with the indicated percentages. **e** 16q isochromosome formation was detected in HeLa b15 (53BP1[hypo]) populations with the indicated percentages. **f** Normal chromosome 16 showing both p-arm and q-arm is maintained in all HeLa cells

## Discussion

Extensive studies report that 53BP1 facilitates repair of double-ended DSBs via the NHEJ pathway in G1, by protecting DNA ends from resection[24–26,35], a key initiation step for HR. During S phase, this activity is neutralised by BRCA1-CtIP, which channels it to an error-free HR repair pathway[40]. In the current study, we have identified that a new role of 53BP1, shown in HeLa and U2OS human cancer cells, is to limit the formation of illegitimate sister DNA entanglements, which otherwise interferes with proper chromosome segregation. The cultured cancer cells,

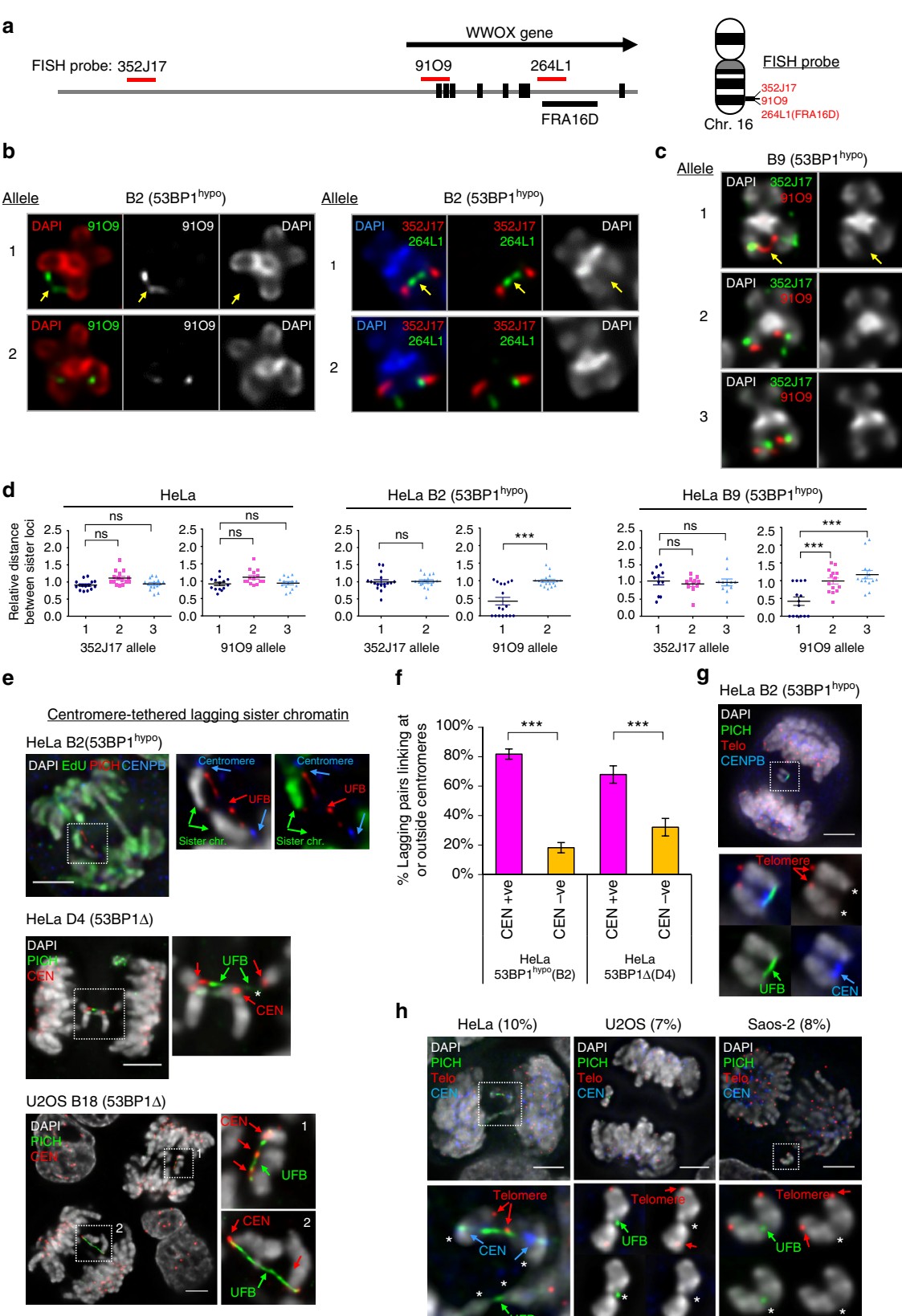

however, not normal diploid cells, are more susceptible to the loss of 53BP1 activities, even when NHEJ function is not fully compromised. This may reflect that 53BP1 has a cell-type specific function. Alternatively, cancerous cell lines may bear high genome instability, DNA replication demands and/or recombination activities, which predispose the formation or accumulation of sister DNA intertwining molecules when 53BP1 activity is limited. Indeed, transformed cells—including HeLa—are reported to display >3-fold increase in HR activity[41]. A striking consequence of 53BP1 depletion in these cancer cells is the elevation of UFB formation that is not associated with the FANCD2 protein (a marker of replication stress or DNA crosslinking). Interestingly, the DNA intertwinements are still found at a common fragile site, but this is not coupled to chromosomal fragility (the latter being a feature of replication stress induced by DNA polymerase inhibition or DNA repair deficiency[42–46]). The lack of fragility at these sites suggest that the new form of UFB caused by 53BP1-deficiency in the cancer cells, may be fundamentally different from those originating from LRIs. The fact that depleting RAD51 can significantly diminish the FANCD2-negative UFBs makes us believe that they are likely a by-product of a HR reaction. However, we cannot rule out that there is a distinct form of replication intermediate structure that does not trigger the FA pathway and chromosomal fragility, and then again retains the intertwining of sister chromatids.

It remains unclear how 53BP1 prevents illegitimate formation of sister-chromatid bridging in the examined cancer cells. It is plausible that (partial) loss of 53BP1 function may lead to the formation of a distinct type of replication intermediate that is subsequently converted by RAD51 into FANCD2-negative DNA intertwining molecules, to prevent fork stalling or incomplete replication. Alternatively, 53BP1 may act to suppress HR initiation and/or promote resolution of HR-mediated joint molecules during DNA replication (Supplementary Fig. 12). Extensive studies have shown that 53BP1 exerts DNA end-blocking, but is counteracted by BRAC1-CtIP during S phase[24–26,40]. It is conceivable that the (partial) loss of 53BP1 activity may relieve the constraints of HR initiation at damaged replication forks. On the other hand, loss of 53BP1 may weaken the anti-recombinogenic and/or dHJ dissolution activities exerted via interaction of the BLM complex[47–49]. Therefore, defects in one or all of these potential activities may result in excessive formation of sister-chromatid bridges.

Another striking finding in the current study is that the new type of sister DNA intertwinement can drive signature chromosome rearrangements, notably, via a distinct chromatid damage process. We termed this as "sister-chromatid rupture-bridging".

Models such as BFB cycle have been proposed to explain the development of gross chromosomal rearrangements via single or multiple rounds of DNA damage introduced, mostly on chromatid arms concomitant with the breakage of anaphase bridges during or after cytokinesis. Subsequent fusion of the broken arms leads to re-generation of anaphase bridges and further breakages in the next cell cycle[7,8]. Contrary to this mechanism, we found that the sister DNA intertwinements induce chromatid rupture upon anaphase onset, and unexpectedly it happens prior to the breakage of the DNA bridges. Consequently, the ruptured sister arms remained tethered and gave rise to the characteristic non-disjunction chromatid products. When occurring at centromeres, it can drive co-segregation of the ruptured whole-arm chromatin and the formation of signature chromosomal rearrangements, including whole-arm (Robertsonian-like) deletions/translocations and isochromosome formation that are as-yet-unexplained alterations observed in tumour cells[50]. Our findings thus provide an alternative explanation for how distinct whole-arm rearrangements may arise from illegitimate sister DNA bridging. It is conceivable that if the rupture-bridging phenomenon occurs on rDNA-bearing chromosomes in germ cells, this could lead to Robertsonian translocations that are present in a subset of Patau and Down syndrome patients[51].

Lagging chromosomes are frequently observed in tumour cells. Based on our in-depth cytogenetic analyses, we speculate that some of the reported lagging chromosome formation may not be due to kinetochore-microtubule attachment errors, but due to the persistence of sister-chromatid bridges presented here. Further examinations will need to revisit the origin(s) of chromosome missegregation in cancers.

CIN is generally thought to be beneficial for tumour progression[1,2]. A recent study has shown that whole-arm deletions are positively correlated with loss of tumour suppressor islands that may confer growth advantages[52]. Undoubtedly, 53BP1 serves as a genome stability guardian and suppresses tumourigenesis as shown in mouse studies[53,54]. However, our study also indicates that in the examined cancer cells, 53BP1 is required to prevent excessive chromosome missegregation and probably genome hyper-instability, and also for optimal growth. Thus, we believe that chromosomal (hyper-)instability may need to be restrained in cancers, (e.g. by 53BP1-mediated pathway) otherwise the adverse effects such as chromatid intertwining and unwanted rearrangements may hinder tumour survival fitness. Thus, targeting the 53BP1 pathway may be, on the other hand, a promising therapeutic remedy in cancer treatments.

In conclusion, we show a distinct mitotic chromatid rupture-bridging process mediated by ultrafine sister DNA

**Fig. 7** Sister-chromatid rupture-bridging is strongly linked to distinct chromosomal rearrangements. **a** Positions of FISH probes at WWOX gene locus on chromosome 16. **b** Representative FISH images showing DNA thread structures linking the promoter region (left) and at CFS-FRA16D site (right) of WWOX sister alleles on HeLa B2 (53BP1$^{hypo}$) metaphase chromosomes. **c** DNA thread structures (91O9 probe; red) were also detected in HeLa b9 (53BP1$^{hypo}$) cells. Arrows indicate DNA threads linking the well-separated chromatid arms. Probe 352J17 was used as a control. **d** Relative distance between sister signals of FISH probes, 352J17 and 91O9, in HeLa and 53BP1$^{hypo}$ cells. FISH signals showing a line or connected dot is considered as zero distance—DNA thread formation. Eighteen metaphase spreads were counted. Note: HeLa B2 (53BP1$^{hypo}$) cells retain only two intact WWOX alleles. **e** Examples of centromere-tethered lagging sister chromatin in 53BP1-depleted HeLa and U2OS cells. A pair of lagging sister chromatin as differentially labelled by EdU (green), intertwined by a PICH-UFB (red) at their centromeres (CENB, blue) in HeLa B2 (53BP1$^{hypo}$) anaphase cell (Top). Pairs of broken lagging chromatin tethered at centromeres (red) by PICH-UFBs (green) in HeLa D4 (53BP1Δ) (Middle) and U2OS B18 (53BP1Δ) cells (Bottom). **f** Frequencies of lagging-chromatin pairs with UFBs linking at centromeres in HeLa 53BP1$^{hypo}$ and 53BP1Δ cells. Numbers of lagging chromatin pairs analysed, B2 = 49 and D4 = 44 from three independent experiments. **g** Immuno-FISH analysis revealed loss of whole chromatid arms on lagging chromatid pairs tethered by UFBs at centromeres. A representative image of HeLa 53BP1$^{hypo}$ cell showing a PICH-UFB (green) intertwines the sister centromeres (blue) of a pair of lagging chromatin, at where the telomeres (red) are missing (asterisks). **h** Frequencies of sister-chromatid rupture-bridging phenomenon in unperturbed HeLa (4/40; 10%), U2OS (3/43; 7%) and Saos-2 (2/25; 8%) cells. Representative images showing ruptured chromatin tethered by PICH-UFBs (green), sometimes at centromeres (blue). Asterisks mark the ruptured positions at where the rest of chromatids is lost, as determined by telomere FISH (red). Scale bars, 5 μm. Error bars represent s.d. of three independent experiments. Statistical significance was determined by T-test (***, p < 0.001; ns, nonsignificant)

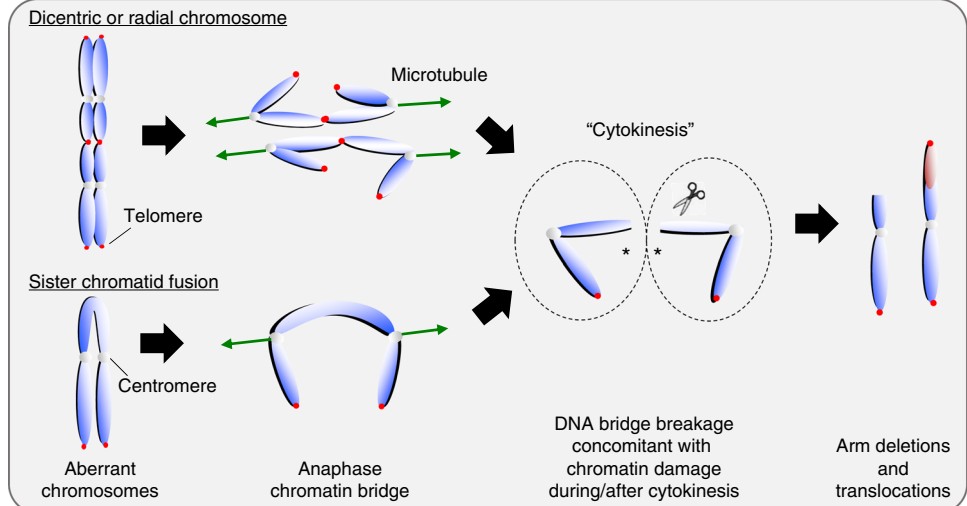

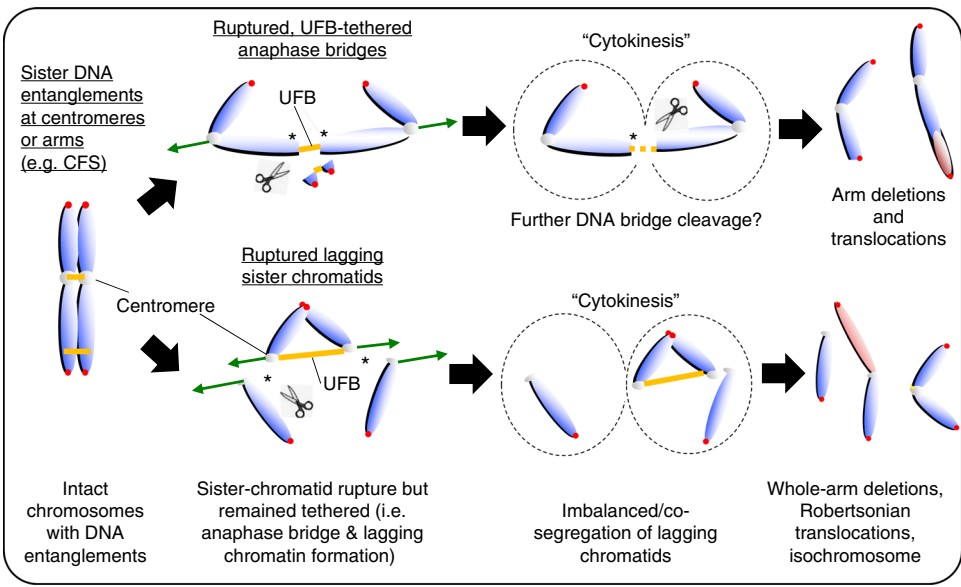

**Fig. 8** Models of gross chromosomal rearrangements driven by conventional anaphase bridge-breakage and sister-chromatid rupture-bridging pathways. **a** Conventional anaphase bridge-breakage (also known as breakage-fusion-bridge cycle) model driven gross chromosomal rearrangements (GCRs). Aberrant chromatids or chromosomes, such as dicentric/radial chromosomes and sister-arm fusion lead to chromatin bridge formation in anaphase. The breakage of anaphase bridges during or after cytokinesis results in chromosome damage, which subsequently can lead to deletions, translocations and/or the re-formation of dicentric chromosomes and sister-arm fusion. Cells enter another anaphase bridge-breakage cycle in the next mitosis that accumulate further chromosome alterations. **b** Sister-chromatid rupture-bridging model driven GCRs (in the current study). Illegitimate formation/accumulation of ultrafine sister DNA intertwinements lead to a symmetrical rupture of sister-chromatid axes (asterisks). The resulting sister arms remain tethered by UFB structures resulting in anaphase bridges or lagging chromatin pairs formation (when the rupture occurs at centromeres). Further breakage may occur on the UFB-tethered anaphase bridges in late mitosis (e.g. during abscission in cytokinesis), which lead to arm deletions or translocations. On the other hand, the centromere-tethered lagging chromatin pairs, which lose the entire opposite arms, may escape abscission and co-segregate into one of the daughter cells. Hence, this provides an ideal precursor for isochromosome formation, or causes whole-arm translocations in the next cell cycle. Our current model, therefore, provides an alternative explanation on the formation of whole-arm rearrangements that may arise in cancer karyotypes

intertwinements that promotes characteristic chromosomal rearrangements in 53BP1-depleted human cancer cells.

## Methods

**Cell culture, treatment and transfection.** All cell lines were obtained from Cell Bank of Genome Damage and Stability Centre and were originally purchased from ATCC. All cell lines were authenticated by STR genotyping from European Collection of Cell Cultures and passed mycoplasma tests (Lonza Mycoplasma testing kit). HeLa, U2OS and SAOS-2 cells were grown in DMEM containing 10% foetal bovine serum (Gibco), L-glutamine and Pen/Strep antibiotics. RPE1-hTERT cells were grown in DMEM/F-12 containing 10% foetal bovine serum (Gibco) and Pen/Strep antibiotics. HCT116 cells were grown in McCoy's 5 A containing 10% foetal bovine serum (Gibco), L-glutamine and Pen/Strep antibiotics. HeLa B2G53BP1 cells were maintained with 0.4 mg/ml G418. Cell cultures were maintained at 37 °C in a humidified atmosphere containing 5% $CO_2$. As indicated, the

cells were treated with the DNA polymerase inhibitor aphidicolin (Sigma, A0781; 0.3 µM).

*53BP1*-knockout HeLa and RPE1 cells were generated using the CRISPR-Cas9 m, with the following guide-RNAs targeting *exon 2* (CAGGTTCTAGAGGATGATTCTGG) for 53BP1^hypo^ and *exon 10* (TTTATC GTTCCTAGCAGTCC) for *53BP1Δ knockout*.

Briefly, gene-specific gRNAs were cloned in pSpCas9(BB)-2A-Puro (Addgene) containing a puromycin resistance cassette. HeLa cells were transfected (Fugene HD, Promega) and RPE1 cells were electroporated (Neon Transfection System, ThermoFisher) with pSpCas9(BB)-2A-Puro-53BP1gRNA. Transfected HeLa, U2OS and RPE1 cells were selected by 0.25 µg/ml and 2 µg/ml of puromycin, respectively, for 72 h. B2 (HeLa-53BP1^hypo^) cells were generated by cotransfecting pMACSK^K^II and pSpCas9(BB)-2A-Puro-53BP1gRNA-b in HeLa cells. Forty-eight hours later, the cells were labelled by MACSselectK^K^II microbeads and enriched by MACSselect K^K^II column without any antibiotic selection.

Individually isolated cells were isolated and screened by immunoblotting and immunofluorescence. The presence of mutations in knockout cells were identified by Sanger sequencing following TA-cloning (35 clones) using primers (Fwd: CAGGATTGGACACAACATCCTAG; Rev: CTCTCAGCAAGATACTCCTTG CC). Primers used for 53BP1 wild-type allele-specific PCR (Fwd; AAGCCAGGT TCTAGAGGATG; Rev: CTCTCAGCAAGATACTCCTTGCC).

To generate B2G53BP1 cells, full-length pEGFP-C3-53BP1 (isoform1) construct was transfected in B2 (53BP1^hypo^) cells and were selected by G418 (0.8 mg/ml) for 2 weeks. Individual clones were screened and cultured.

HeLa or B2 cells were transfected with siRNAs using Lipofectamine RNAiMAX (Invitrogen) following the manufacturer's instruction. The sequences of stealth siRNAs (ThermoFisher) are as follows:

RAD51 (CCACCAGACCCAGCUCCUUUAUCAA),

Non-targeting pool (UGGUUUACAUGUCGACUAA,UGGUUUACAUGU UGUGUGA, UGGUUUACAUGUUUUCUGA and UGGUUUACAUGUUU UCCUA).

**Fluorescence immunostaining**. For immunostaining analyses, the cells were seeded onto No:1.5 H cover glass and fixed with PFA buffer (250 mM HEPES, 1x PBS, pH7.4, 0.1% Triton X-100, 4% methanol-free paraformaldehyde) for 20 min at 4 °C or with room temperature PFA buffer (1x PBS, 4% methanol-free paraformaldehyde) for 10 min. Pre-extraction was carried out in indicated experiments before fixation by incubation of the cover glass into pre-extraction buffer (20 mM HEPES, pH 7.4, 0.5% Triton X-100, 50 mM NaCl, 3 mM MgCl$_2$, 300 mM sucrose) for 15 sec.

Primary antibodies used: anti-FANCD2 (Novusbio NB100-82; 1:600), anti-53BP1 (Abcam ab36823; 1:800), anti-53BP1 (Bethyl Lab A300-272A; 1:1000), anti-53BP1 (Millipore MAB3802; 1:800), anti-53BP1 (Santa Cruz H-300; 1:400), anti-RIF1 (Bethyl Lab A300-568A-3; 1:200), anti-PICH (Abnova H00054821-B01P; 1:150), anti-PICH (Abnova; H00054821-D01P; 1:100), anti-γH2AX (Upstate JBW-301; 1:400), anti-RAD51 (Abcam ab63801; 1:200), anti-RPA70 (Abcam ab79398; 1:200), anti-CENPB (Abcam ab25734; 1:600) and anti-centromere (ImmunoVision HCT-0100; 1:800). All secondary antibodies were used in a dilution of 1:500 and purchased from ThermoFisher Scientific, unless stated otherwise. Secondary antibodies used: donkey anti-mouse Alexa Fluor 488 (A-21202), 555 (A-31570) and 647 (A-31571); donkey anti-rabbit Alexa Fluor 488 (A-21206), 555 (A-31572) and 647 (A-31573); donkey anti-goat Alexa Fluor 555 (A-21432); goat anti-rabbit Abberior STAR 635 P (Abberior 2-0012-007-2) and goat anti-human DyLight 550 (Abcam ab96908) and 650 (Abcam ab96910). Immunofluorescence staining was performed according to previously described protocols[13]. In brief, the samples were incubated with primary antibodies at 37 °C for 90 minutes and rinsed with 1 × PBS followed by incubation of secondary antibodies in room temperature for 25 minutes. The cells were mounted using Vectasheild containing DAPI.

**High-resolution deconvolution microscopy**. Image acquisition was carried out under a Zeiss AxioObserver Z1 epifluorescence microscopy system with 40 × /1.3 oil Plan-Apochromat, 63 × /1.4 oil Plan-Apochromat and 100 × /1.4 oil Plan-Apochromat objectives and a Hamamatsu ORCA-Flash4.0 LT camera. The system is calibrated and aligned by using 200 nm-diameter TetraSpeck microspheres (ThermoFisher). Z-stack images were acquired at 0.2 µm intervals covering a range from 3–8 µm by using ZEN blue software.

Deconvolution was carried out using Huygens Professional deconvolution software (SVI) with a measured point-spread-function generated by 200 nm-diameter TetraSpeck microspheres. Classical maximum likelihood estimation method with iterations of 40–60 and signal-to-noise of 20–40 was applied.

**Anaphase bridges and lagging chromosomes counting**. Cells were grown on coverslips for 18 h and fixed with 4% paraformaldehyde in 1x PBS at room temperature for 10 min. The cells were then stained with Hoechst 33342 and slides were mounted in Vectashield mounting medium. Anaphase cells with chromatin bridges and lagging chromatin were scored under Zeiss Axio Observer Z1 microscope. The counting was completed by three different assessors between experiments. The cells were counted from random microscopic fields.

**Differential sister-chromatid labelling on mitotic cells**. Differential labelling of sister chromatids was performed by modifying the cell synchronisation method of a double thymidine arrest. Briefly, the cells were incubated with EdU during the second thymidine arrest period. EdU was then maintained following the release and washed away after 9 h of incubation. After 24–26 h, the samples were collected for metaphase spread preparation or anaphase cell fixation as mentioned before and subjected to immunofluorescence staining and microscopy analysis.

EdU was detected using Click-iT Plus EdU labelling kits (Alexa Fluor 488, 555 or 647). For EDU pulse labelling, the cells were treated with EdU (10 µM) 10–15 min prior to fixation. EdU staining was performed according to the manufacturer's instruction.

**Immunoblotting**. Cells were trypsinized and lysed on ice for 20 min with lysis buffer (50 mM Tris, pH 7.5, 300 mM Nacl, 5 mM EDTA, 1% Triton X-100, 1.25 mM DTT, 1 mM PMSF and cOmplete^TM^ protease inhibitor cocktail). Protein concentration was quantified using Bradford assay (Bio-Rad). Immunoblotting (IB) was performed following standard procedures. Primary antibodies used for IB in this study: anti-53BP1 (Abcam, ab36823; 1:7000), anti-53BP1 (Santa Cruz, H-300; 1:800), anti-Ku80 (Abcam, ab80592; 1:10000) and anti-β-actin (Sigma, A5316; 1:5000).

Secondary antibodies used: goat anti-rabbit HRP (Amersham NA9340V; 1:25000) and goat anti-mouse HRP (Abcam ab6789; 1:10000).

**MTT proliferation assay**. Cell proliferation was assessed using MTT (3-(4,5-dimethylthiazol-2-yl)-2,5-diphenyltetrazolium bromide). Briefly, the cells were seeded in triplicate on a 24-well plate and 50 µl of 0.5 g/L MTT was added at indicated times. After incubation at 37 °C for 2 h, the MTT medium was aspirated and 250 µl of DMSO was added to each well. The absorbance was measured using CLARIOstar plate reader (BMG Labtech) at 595 nm.

**Clonogenic cell survival and Micro-colony formation assay**. For clonogenic cell survival assay, HeLa, U2OS and its derived 53BP1 knockout cells or RPE1 and its derived 53BP1 knockout cells were plated in 10-cm tissue culture petri dishes. Five hours later, the cells were treated with different doses of irradiation using an X-ray machine. The treated cells were allowed to grow for 15 days to form colonies. The colonies were fixed with 70% ethanol and stained with 1 mg/ml bromophenol blue for 2 h.

For micro-colony formation assay, ~50 single cells were seeded onto a coverslip (24 × 24 mm) to form colonies. After 7 days, the cells were stained with CellMask™ Deep Red (ThermoFisher Scientific) and fixed with PFA. The cells were mounted in DAPI Vectasheild and captured under a 40 × objective using 3 × 3 tiling to image the whole colony size in an area of 900 × 900 µm$^2$.

**Metaphase spread preparation**. Cells were collected for metaphase spread preparation after 1 h of colcemid (Gibco, 0.5 µg/ml) treatment and were swelled with pre-warmed KCl (0.075 M) hypotonic solution at 37 °C for 5 min. The cells were washed twice and fixed with 3:1 methanol:acetic acid. The cells were dropped onto glass slides. For chromosome number frequency the metaphase spreads from random microscopic fields were counted.

**Telomere end-to-end fusion assay**. For telomere fusion assay pLPC-NMYC TRF2^ΔBΔM^ construct (Addgene) was transiently transfected in HeLa or B2 (53BP1^hypo^) for 36 h using Fugene HD. Following transfection, the cells were harvested for metaphase spread preparation and quantitation after FISH staining (described below).

**Fluorescence in situ hybridisation (FISH)**. Bacterial artificial chromosome (BAC) clones were purchased from BACPAC resources centre, C.H.O.R.I. BAC DNA was isolated using QIAGEN® Plasmid Purification MaxiPrep kit. A volume of 1 µg of BAC clone DNA was labelled by nick translation for 90 min at 15 °C with digoxigenin-11-dUTP (Roche) using a nick translation kit (Roche) or ATTO labelling kit (Jena Bioscience). Labelled probe together with cot-1 DNA (Roche) was dehydrated. The denatured probe was re-suspended in hybridisation buffer and placed on a metaphase slide (prepared as described earlier), which had been dehydrated in a graded ethanol series (70%, 85% and 100%). The slide was sealed, denatured (82.5 °C for 2 min) and incubated O/N at 37 °C. The slide was washed at 65 °C in 0.1 × SSC. DIG-labelled probe was detected with FITC-conjugated anti-digoxigenin for 30 min at RT. Chromosomes were counterstained with DAPI Vectasheild.

**Peptide nucleic acid (PNA) fluorescent in situ hybridisation**. PNA probes were hybridised according to the manufacturer's instructions telomere-CY3 (DAKO, Agilent technologies) or FAM488-CENPB (PNAbio). Briefly, metaphase slides were washed in TBS buffer, fixed in 3.7% PFA and dehydrated in a graded ethanol series (70%, 85% and 100%). The slides were air dried and hybridised with PNA probe and co-denatured (80 °C for 2 min) and incubated for 2 h at room temperature. Chromosomes were counterstained with DAPI Vectasheild.

**Immuno-FISH**. Immuno-FISH was performed after standard IF-staining procedures. Briefly, the cells were subjected to immunofluorescence as described above and re-fixed with 8% paraformaldehyde at room temperature for 10 min. The samples were co-denatured with 10 µl of Cy-3-labelled telomere specific PNA probe (80 °C for 5 mins) and hybridised for 2 h at room temperature in the dark. The cells were then washed and mounted with DAPI Vectasheild.

**Flow cytometry**. The cells were trypsinised, washed with PBS and fixed with 70% ice-cold ethanol. For cell cycle analysis, the cells were washed with PBS and re-suspended in PI/RNase staining buffer. FACS profile were then determined and analysed using BD accuri C6 sampler.

**Statistics**. Statistical analysis was performed using GraphPad Prism software by two-tailed unpaired Student's $t$-test or two-way Anova as per the experimental requirement. Data were presented as the mean + s.d. unless specified. Probability value '$p \leq 0.05$' was considered to be significant.

**Data availability**. All the data and materials supporting this work are available upon reasonable request to the corresponding author.

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

## Acknowledgements

We thank people in the Genome Centre for their great support and help. We would also like to thank Nadia Hegarat for puromycin-sensitive RPE1-hTERT cells and Matthew Neale, Aidan Doherty and Anthony Carr for helpful comments on the manuscript. We also thank Stephen West for sharing unpublished research information. This work is supported by Sir Henry Dale Fellowship (Ref: 104178/Z/14/Z) from Wellcome Trust and the Royal Society, and by the Genome Damage and Stability Centre. K-L.C. is the recipient of Sir Henry Dale Fellowship. Funding for open access charge: Charity Open Access Fund (COAF).

## Author contributions

A.T. and K.-L.C. designed and performed the experiments with help from O.A.J. A.T. O.A.J. and K.-L.C. wrote the manuscript.

## Additional information

**Competing interests:** The authors declare no competing financial interests.

