## [Peer Review File · Nature Communications]

Reviewers' Comments:

Reviewer #1:

Remarks to the Author:

In the manuscript entitled "53BP1 limits recombination-mediated sister chromatid rupture and rearrangements in human cancer cells driven by a distinct DNA bridge-breakage pathway" by Tiwari and Colleagues, the authors claim that depletion of the 53BP1 protein in HeLa and U2OS cells, but not RPE1 cells exacerbates chromatin non-disjunction due to chromatin entanglements resulting from homologous recombination activity.

In my opinion, the claim that the chromosome non-disjunction observed is specific for cancer cells is not well supported, as only HeLa and U2OS are compared to a single h-TERT immortalised cell line, RPE1, where no disjunction issues were observed. To support the statement that these mitotic defects are a general feature of cancer cells a larger panel of cancer cell lines depleted for 53BP1, as well as a panel of human primary cells depleted for this protein should be examined. As it stands this claim is not well supported and should either be investigated further or re-worded within the text.

Specific comments:

In their rebuttal, the authors claim that "We believe that our data has clearly shown that this new type of UFBs is distinct from the one generated under replication stress. First, the HR-UFB is not associated with FANCD2, a key marker for replication stress structure in mitosis, but instead is positive for γ H2AX. Second, the new UFBs arise from intact sister chromatids whereas the replication stress-induced one is often associated with chromosome fragility. Third, the formation of the new UFBs, but not the replication intermediate one, is dependent on RAD51. Based on these findings, we thus classify the new sister DNA bridges as a third type of UFBs."

While I agree that the authors present some new evidence that may support the idea that the UFBs generated by the depletion of 53BP1 and RAD51 constitute a novel class of UFB that arise from HR intermediates, there are some essential controls and analyses missing from the data presented. Specifically, in figure 4a and b the frequency of PICH coated UFBs associated with FANCD2 is presented in U2OS and HeLa 53BP1 knockout cells compared to HeLa 53BP1 hypo cells and hypo cells treated with aphidicolin. There are no statistical analyses indicated when comparing these cells, and crucially the essential controls of the parental HeLa and U2OS cells in the presence or absence of aphidicolin, as well as aphidicolin-treated knockout cell lines are missing and should be included. Without the statistical analysis and these essential controls one cannot support the statement that "the UFBs arising from 53BP1 depletion are not associated with FANCD2". The statement that these novel UFBs are associated with γ H2AX is also not well supported as only staining in cells depleted for 53BP1 (Fig 4e) is presented. Parental control cells should also be analysed and compared to the depleted cells for the frequency of γ H2AX associated UFBs.

In response to my original comment "The authors state that the 53BP1 hypo HeLa cells display a slow growth phenotype and chromosome non-disjunction defects which were not observed in the RPE1 hypo cells. Why is this? Is it possible that further depletion of 53bp1 in these cells will result in such a phenotype?"

The authors state "we find the defects seem to be cancer cells dependent. RPE1 53BP1 Δ cells do not exhibit severe mitotic and growth defects. However, to strengthen our study we have now included the analysis of 53BP1 Δ HeLa and U2OS cells (new Fig 2 and Supplementary 4)."

However, as mentioned above, the inclusion of only two cancer cell lines vs RPE1 cells as a comparison is not sufficient evidence to claim that the phenotypes observed are specific to cancer cells. This claim should be supported by analysis of primary cells or removed/amended from future versions of the manuscript.

In response to my question "Do drugs that induce DSBs which require HR-mediated repair cause an increase in these UFBs? For example, drugs that collapse replication forks during S-Phase?" the authors respond that "Any drugs that interfere the DNA replication could potentially increase

the replication intermediate-related UFBs. Since, we have observed the formation of HR-mediated bridges without any exogenous DNA assaults, to avoid the complexity of the additional DNA damage and unwanted side effects due to its usage. However, a crucial experiment in our study is that we can suppress the formation of the FANCD2-negative sister DNA bridges by RAD51 knockdown, suggesting their generation depends on the HR pathway (new Fig. 4 f-i)."

I do agree in principle that demonstrating that RAD51 knockdown suppresses the formation of FANCD2-negative UFBs is good evidence that these UFBs originate from HR intermediates. However, as mentioned above, the frequency of FANCD2-associated UFBs in the absence of 53BP1 is not quantified using the appropriate controls and importantly the statistical analyses are missing (as such this precludes any conclusion). Thus, these analysis must be included to lend validity to the claim that 53BP1 depletion leads to the formation of a novel class of UFBs.

Reviewer #2:

Remarks to the Author:

In their revised manuscript,

Tiwari and colleagues have made changes to their original manuscript, and have addressed some of my concerns. I have to say that I find some of the rebuttal comments useful and on-point, whereas some other comments do not reply to concerns, but rather highlight (unrelated) experiments.

Overall, multiple experiments initially lacked sufficient amounts of replicates and statistics. This issue has been addressed throughout the paper, except for a few examples. In the bar graphs for Figure 1F, some bars have n=3, whereas other have n=2, and statistics are only present for some of the data for that reason. Figure 7D also only has partial statistics. Figure 7F has statistics, but it is unclear which conditions have been compared to reach these P values. Figure 7 lacks quantification in the figure. Numbers (without statistics) are mentioned in the results section, but should also be present in the figures (with statistics).

I am pleased with the additional experiments using 53BP1-deleted U2OS and HeLa cell lines, which show that the observed phenotype is not restricted to HeLa cells with a hypomorph 53BP1 allele.

I am still not convinced that these HR-dependent chromosomal intertwinings actually underlie the structural aberrancies observed in cancer, and I find conclusions in the text at multiple occasions over-interpreted: 'Our study thus provides a novel insight into the development of complex karyotypes within cancers'. It is not proven that this pathway drives complex karyotypes in cancers. Actually, the 'rupture' phenotype in the SAOS, U2OS and HeLa not necessarily represents two separate sisters connected with a UFB. PICH also localizes to kinetochores, and only is recruited to UFB, when DNA is under tension. It is difficult to imagine how these ruptured structures can still be under tension, and how UFBs can be visualized with PICH staining.

Minor comments:

-Figure 1F: right-most panel, Y-axis: 'spred' should be 'spread'.

-Some parts of the text may require grammar correction and layout check.

-Throughout the methods section, symbols are replaced with squares. Difficult to indicate exact places, since page numbers are lacking.

-Model: figure 8, why can't the model be the same for other replication-based UFBs?

-Figure 5H: the HeLa hypomorph cell lines is rescued with a GFP-53BP1 cDNA. In the image, PICH is stated to be in green. How can this green signal be discriminated from the 53BP1 signal?

Reviewer #3:

Remarks to the Author:

This is a review of NCOMMS-17-25192-T. I was not involved in the first round of review, and so while I am reviewing the manuscript as a whole, I will focus mostly on the concerns in the first review, and the responses. The goal of this manuscript is to define the anaphase chromosome separation defects in cells deficient for the DNA damage response factor 53BP1, examine details of these events that may provide insight into mechanism, and provide measurements of such defects at a known fragile site locus, using FISH. Understanding the sources of mitotic separation defects is a key question in the field of genome maintenance and cancer biology. I agree with the first reviewers that overall the description of anaphase abnormalities in 53BP1-deficient cells is interesting, as is the sister chromatid labeling approach / fragile site thread detection in Fig 7. Regarding what I see as the key concerns from the first review:

1. The reviewers suggested including a 53BP1 knockout line, in part because the mechanism of action of the 53BP1-hypomorphic allele is poorly understood. This 53BP1-KO analysis has been included for most experiments and largely phenocopies the hypomorph, and as such has been adequately addressed.

2. The reviewers were concerned that the phenotypes were only studied in HeLa cells, and hence it is unclear if the phenomenon are general. This is particularly important since HeLa and U2OS cells already show a very high level of anaphase bridges / ultra-fine bridges (UFB) in the control treatments. In response the authors provide experiments in RPE immortalized cells, and find that 53BP1 has no effect on these cells. On the one hand, these experiments address the concerns of the reviewers, but the findings support their concern that the influence of 53BP1 on limiting anaphase aberrations may have limited significance outside HeLa and U2OS cells. However, in my opinion, the fact that RPE cells are included has improved the manuscript, which will allow the readership to evaluate the significance for themselves. So, I find this concern adequately addressed.

3. A major concern of the reviewers were in the interpretation of the UFBs in 53BP1-deficient cells as "recombination-mediated." Specifically, the concern that this conclusion is indirect. The response has largely been to further argue the point. I follow their argument: they find that while 53BP1- cells show predominantly FANCD2- UFBs, siRAD51 show cells with FANCD2+ UFBs (and reduces FANCD2- UFBs). Thus, since the FANCD2- UFBs are reduced in siRAD51 cells, the UFBs caused by 53BP1 loss are "HR-mediated UFBs." While logical, there are many problems with this conclusion. 1) The control cells also show predominantly FANCD2- UFBs, so it is unclear that 53BP1 has anything to do with this. 2) It is unclear what FANCD2 localization is reflecting in this experiment. Certainly, one possibility is that 53BP1 suppresses UFBs that are mediated by RAD51. However, a challenge to this interpretation is that both 53BP1 loss and RAD51 disruption cause elevated UFBs, which makes it difficult to ascribe the order of events. For example, another possibility is that 53BP1 be acting upstream to increase replication stress, and then RAD51 is important to convert such events into FANCD2-. Similarly, 53BP1 could be causing a different type of replication stress compared to Aphidicolin, where again 53BP1 could be acting upstream, and then APH could convert such events to FANCD2- intermediates. The contradiction is similar to that observed with loss of BLM and RAD51, where both cause an increase in UFBs, but only the former causes elevated sister chromatid exchanges (i.e. elevated SCEs don't necessarily mean more HR, but can reflect elevated crossing over). Plus, there is data in the literature supporting a role of 53BP1 in mediating appropriate HR, which contradicts their assertion of a simple HR-inhibition role of 53BP1. The bottom line here is that I agree with the prior reviewers that the authors have over-interpreted the mechanistic insight in their findings, which in fact may limit the overall impact of

the work in the long run. I suggest that the authors focus on the interesting observation that 53BP1 loss causes a distinct subset of UFBs compared to APH treatment or RAD51 depletion, and then provide an even-handed discussion of possible mechanistic explanations, which includes reworking the Title, etc...

We would like to thank all reviewers for their helpful comments. We have now amended the text according to their suggestions, and also provide additional data to support our study.

Reviewers' comments:

Reviewer #1 (Remarks to the Author):

In the manuscript entitled "53BP1 limits recombination-mediated sister chromatid rupture and rearrangements in human cancer cells driven by a distinct DNA bridge-breakage pathway" by Tiwari and Colleagues, the authors claim that depletion of the 53BP1 protein in HeLa and U2OS cells, but not RPE1 cells exacerbates chromatin non-disjunction due to chromatin entanglements resulting from homologous recombination activity.

In my opinion, the claim that the chromosome non-disjunction observed is specific for cancer cells is not well supported, as only HeLa and U2OS are compared to a single h-TERT immortalised cell line, RPE1, where no disjunction issues were observed. To support the statement that these mitotic defects are a general feature of cancer cells a larger panel of cancer cell lines depleted for 53BP1, as well as a panel of human primary cells depleted for this protein should be examined. As it stands this claim is not well supported and should either be investigated further or re-worded within the text.

Our Response:

We understand the reviewer's concern and in the revised manuscript, we have removed the claim of cancer cell specific and have re-worded the text to state that the phenotypes (sister chromatid bridging and slow growth) are observed in HeLa and U2OS cancer cell lines. This new phenomenon could be cell type specific or cancer cell related (revised manuscript pages 3, 5 & 12). Thus, in the present study, we focus to characterise the origins and the consequences of sister DNA bridging on chromosome rearrangements after 53BP1 depletion.

Specific comments:

In their rebuttal, the authors claim that "We believe that our data has clearly shown that this new type of UFBs is distinct from the one generated under replication stress. First, the HR-UFB is not associated with FANCD2, a key marker for replication stress structure in mitosis, but instead is positive for γ H2AX. Second, the new UFBs arise from intact sister chromatids whereas the replication stress-induced one is often associated with chromosome fragility. Third, the formation of the new UFBs, but not the replication intermediate one, is dependent on RAD51. Based on these findings, we thus classify the new sister DNA bridges as a third type of UFBs."

While I agree that the authors present some new evidence that may support the idea that the UFBs generated by the depletion of 53BP1 and RAD51 constitute a novel class of UFB that arise from HR intermediates, there are some essential controls and analyses missing from the data presented. Specifically, in figure 4a and b the frequency of PICH coated UFBs associated with FANCD2 is presented in U2OS and HeLa 53BP1 knockout cells compared to HeLa 53BP1 hypo cells and hypo cells treated with aphidicolin. There are no statistical analyses indicated when comparing these cells, and

crucially the essential controls of the parental HeLa and U2OS cells in the presence or absence of aphidicolin, as well as aphidicolin-treated knockout cell lines are missing and should be included. Without the statistical analysis and these essential controls one cannot support the statement that “the UFBs arising from 53BP1 depletion are not associated with FANCD2”.

Our Response:

We understand the reviewer’s concern. In the revised manuscript, we provide additional data in new Figure 4a to show that 53BP1 depletion in HeLa and U2OS cells significantly increases anaphases having FANCD2-negative UFBs as compared to their parental control cell lines. Moreover, we also provide additional data in new Supplementary Figure 7a to show that aphidicolin treatment increases HeLa anaphases having FANCD2-positive, but not FANCD2-negative UFBs. Statistical analysis has been carried out to analyse the significance.

Moreover, consistent to these results, we have shown that 53BP1^{hypo} cells do not exhibit increased formation of mitotic FANCD2 sister foci (Supplementary Fig. 7c-d) but increase the formation of UFBs (Fig. 2e), inferring that the DNA bridges are unlikely associated with FANCD2. Together with the new data, these results reemphasize that 53BP1 suppresses the formation of a type of UFBs negative of FANCD2.

In Fig. 4b (now moved to Supplementary Fig. 7b), we aim to show that a high proportion of UFBs in 53BP1-depleted HeLa and U2OS is negative of FANCD2 and to show this is not due to defects in FANCD2 loading because aphidicolin treatment in 53BP1^{hypo} cells can largely increase the proportion of FANCD2-positive UFBs. Moreover, the aphidicolin treatment also induced FANCD2 sister foci in both HeLa and 53BP1^{hypo} prometaphase cells (Supplementary Fig. 7c). These data indicate that the FA pathway remains active.

The statement that these novel UFBs are associated with gammaH2AX is also not well supported as only staining in cells depleted for 53BP1 (Fig 4e) is presented. Parental control cells should also be analysed and compared to the depleted cells for the frequency of gammaH2AX associated UFBs.

Our Response:

We understand the reviewer’s concern. We would like to clarify that the counting of γ H2AX was done on the “UFB-tethered chromatin bridges”, which is a very distinct mis-segregation feature present in the 53BP1^{hypo} cells (please see Fig. 2a, 2c, 3d-e, 4a). We were interested to explore if they are associated with any DNA damage response protein (e.g. γ H2AX) and found nearly 75% of them are γ H2AX positive. Since the UFB-tethered chromatin bridges are not frequently observed in the parental HeLa cells, it is therefore not practical to compare the γ H2AX frequencies. We have now amended the text in the revised manuscript to clarify this (page 7).

In response to my original comment “The authors state that the 53BP1 hypo HeLa cells display a slow growth phenotype and chromosome non-disjunction defects which were not observed in the RPE1 hypo cells. Why is this? Is it possible that further depletion of 53bp1 in these cells will result in such a phenotype?”

The authors state “we find the defects seem to be cancer cells dependent. RPE1 53BP1 Δ cells do not

exhibit severe mitotic and growth defects. However, to strengthen our study we have now included the analysis of 53BP1 Δ HeLa and U2OS cells (new Fig 2 and Supplementary 4).”

However, as mentioned above, the inclusion of only two cancer cell lines vs RPE1 cells as a comparison is not sufficient evidence to claim that the phenotypes observed are specific to cancer cells. This claim should be supported by analysis of primary cells or removed/amended from future versions of the manuscript.

Our Response:

As per reviewer’s suggestion, in the revised manuscript we have removed the claim of phenomenon to be cancer cell specific, instead we state that HeLa and U2OS cancer cells show a higher reliance on 53BP1 activity (page 5).

In response to my question “Do drugs that induce DSBs which require HR-mediated repair cause an increase in these UFBs? For example, drugs that collapse replication forks during S-Phase?” the authors respond that “Any drugs that interfere the DNA replication could potentially increase the replication intermediate-related UFBs. Since, we have observed the formation of HR-mediated bridges without any exogenous DNA assaults, to avoid the complexity of the additional DNA damage and unwanted side effects due to its usage. However, a crucial experiment in our study is that we can suppress the formation of the FANCD2-negative sister DNA bridges by RAD51 knockdown, suggesting their generation depends on the HR pathway (new Fig. 4 f-i).”

I do agree in principle that demonstrating that RAD51 knockdown suppresses the formation of FANCD2–negative UFBs is good evidence that these UFBs originate from HR intermediates. However, as mentioned above, the frequency of FANCD2-associated UFBs in the absence of 53BP1 is not quantified using the appropriate controls and importantly the statistical analyses are missing (as such this precludes any conclusion). Thus, these analysis must be included to lend validity to the claim that 53BP1 depletion leads to the formation of a novel class of UFBs.

Our Response:

We thank the reviewer for the suggestion. We have now provided the additional data in new Figure 4a and in new Supplementary Figure 7a to show that, unlike aphidicolin treatment, (partial) depletion of 53BP1 in HeLa and U2OS cells can significantly induce FANCD2-negative UFBs and its frequency is significantly reduced after RAD51 knockdown (Fig. 4i).

Reviewer #2 (Remarks to the Author):

In their revised manuscript,

Tiwari and colleagues have made changes to their original manuscript, and have addressed some of my concerns. I have to say that I find some of the rebuttal comments useful and on-point, whereas some other comments do not reply to concerns, but rather highlight (unrelated) experiments.

Overall, multiple experiments initially lacked sufficient amounts of replicates and statistics. This issue has been addressed throughout the paper, except for a few examples. In the bar graphs for Figure 1F, some bars have n=3, whereas other have n=2, and statistics are only present for some of the data for that reason.

Our Response:

We have now repeated the experiments and modified Fig. 1f in the revised manuscript. Statistical analysis has also been carried out to analyse the significance.

Figure 7D also only has partial statistics. Figure 7F has statistics, but it is unclear which conditions have been compared to reach these P values. Figure 7 (Note: 7H?) lacks quantification in the figure. Numbers (without statistics) are mentioned in the results section, but should also be present in the figures (with statistics).

Our Response:

Thanks for pointing it out. We have completed the statistical analysis in Figure 7D and re-plotted Figure 7F in the revised manuscript to show UFB linkage occurred significantly at centromeres of lagging chromatin pairs.

In Figure 7H, we aimed to investigate the presence of non-intact lagging chromatin pairs (not lagging chromosomes) in various cancer cell lines under unperturbed condition. Therefore, there is no comparison. We have put the number and percentage of anaphase cells analysed from each cell lines having non-intact lagging chromatin pairs in Figure 7h legend.

I am pleased with the additional experiments using 53BP1-deleted U2OS and HeLa cell lines, which show that the observed phenotype is not restricted to HeLa cells with a hypomorph 53BP1 allele.

I am still not convinced that these HR-dependent chromosomal intertwinings actually underlie the structural aberrancies observed in cancer, and I find conclusions in the text at multiple occasions over-interpreted: 'Our study thus provides a novel insight into the development of complex karyotypes within cancers'. It is not proven that this pathway drives complex karyotypes in cancers.

Our Response:

Thanks for the suggestion. We have re-worded our text in the revised manuscript (abstract, pages 4 & 12).

Actually, the 'rupture' phenotype in the SAOS, U2OS and HeLa not necessarily represents two separate sisters connected with a UFB. PICH also localizes to kinetochores, and only is recruited to UFB, when DNA is under tension. It is difficult to imagine how these ruptured structures can still be under tension, and how UFBs can be visualized with PICH staining.

Our Response:

In these cases, we interpreted two sister chromatids based on the length/size and the symmetry pattern of DAPI staining on lagging chromatin pairs.

We agree that presumably tension is one of the factors to recruit PICH, probably via changing the conformation of the DNA helix. However, it is highly likely that PICH can remain on the abnormal DNA structures transiently even after the stretched DNA structure is resolved or cleaved. Indeed, we observe for instance in Figure 7e (middle image: HeLa D4), PICH is also found on chromatin mass (top-right area) where no stretched DNA structure is present. Moreover, we do frequently observe PICH foci remaining on chromatin arms, and some of these foci co-localise with FANCD2 in anaphase cells after pre-treatment with aphidicolin (e.g. Figure 4c and the image below). Presumably, PICH can label sites where the DNA entanglements had occurred and/or remain. Therefore, it is not unexpected to see PICH foci at a position potentially linking the lagging chromatin pairs in Fig. 7h.

HeLa anaphase cell pre-treated with aphidicolin showing PICH (green) exhibits as foci, but not stretched structures, with some of FANCD2 (red) foci.

Minor comments:

-Figure 1F: right-most panel, Y-axis: 'spred' should be 'spread'.

Thanks, it has now been corrected.

-Some parts of the text may require grammar correction and layout check.

Thanks, we have gone through the manuscript again and made relevant changes.

-Throughout the methods section, symbols are replaced with squares. Difficult to indicate exact places, since page numbers are lacking.

Thanks, we have corrected the symbols.

Model: figure 8, why can't the model be the same for other replication-based UFBs?

Our Response:

We understand reviewers concern. Indeed, we cannot rule out that replication stress-mediated UFBs may generate a similar chromatid rupture phenomenon. However, this is beyond the scope of our investigation because, for instance, replication stress induced by aphidicolin did not usually generate those distinct UFB-tethered chromatid products (e.g. Figure 2c, 5c and 5f VS Figure 4c), which makes it very difficult to characterise if there is any chromatid rupture in anaphase following replication stress treatments. More importantly, this distinct patterns of mis-segregation again supports the idea that the loss of 53BP1 induces a different form of sister chromatid bridges (FANCD2-negative) as compared to those induced by DNA polymerase inhibition.

-Figure 5H: the HeLa hypomorph cell lines is rescued with a GFP-53BP1 cDNA. In the image, PICH is stated to be in green. How can this green signal be discriminated from the 53BP1 signal?

Our Response:

We understand reviewers concern. Importantly, we are able to perform these analyses because 53BP1 protein does not bind to mitotic chromosomes and DNA bridges during mitosis (except to kinetochores in prometaphase and metaphase cells). 53BP1 also displays a diffused staining pattern (please see Supplementary Figure 5b), so we exploited the green channel to analyse any other protein's localisation in anaphase such as the PICH-stained DNA bridges, which display a defined structure with very high signal-to-noise ratio.

Reviewer #3 (Remarks to the Author):

This is a review of NCOMMS-17-25192-T. I was not involved in the first round of review, and so while I am reviewing the manuscript as a whole, I will focus mostly on the concerns in the first review, and the responses. The goal of this manuscript is to define the anaphase chromosome separation defects in cells deficient for the DNA damage response factor 53BP1, examine details of these events that may provide insight into mechanism, and provide measurements of such defects at a known fragile site locus, using FISH. Understanding the sources of mitotic separation defects is a key question in the field of genome maintenance and cancer biology. I agree with the first reviewers that overall the description of anaphase abnormalities in 53BP1-deficient cells is interesting, as is the sister chromatid labeling approach / fragile site thread detection in Fig 7. Regarding what I see as the key concerns from the first review:

1. The reviewers suggested including a 53BP1 knockout line, in part because the mechanism of action of the 53BP1-hypomorphic allele is poorly understood. This 53BP1-KO analysis has been included for most experiments and largely phenocopies the hypomorph, and as such has been adequately addressed.

We thank the reviewer for the encouraging comment and agreeing that our additional analyses has been adequately addressed.

2. The reviewers were concerned that the phenotypes were only studied in HeLa cells, and hence it is unclear if the phenomenon are general. This is particularly important since HeLa and U2OS cells already show a very high level of anaphase bridges / ultra-fine bridges (UFB) in the control treatments. In response the authors provide experiments in RPE immortalized cells, and find that 53BP1 has no effect on these cells. On the one hand, these experiments address the concerns of the reviewers, but the findings support their concern that the influence of 53BP1 on limiting anaphase aberrations may have limited significance outside HeLa and U2OS cells. However, in my opinion, the fact that RPE cells are included has improved the manuscript, which will allow the readership to evaluate the significance for themselves. So, I find this concern adequately addressed.

We thank the reviewer agreeing that the concern has been addressed.

3. A major concern of the reviewers were in the interpretation of the UFBs in 53BP1-deficient cells as "recombination-mediated." Specifically, the concern that this conclusion is indirect. The response has largely been to further argue the point. I follow their argument: they find that while 53BP1- cells show predominantly FANCD2- UFBs, siRAD51 show cells with FANCD2+ UFBs (and reduces FANCD2- UFBs). Thus, since the FANCD2- UFBs are reduced in siRAD51 cells, the UFBs caused by 53BP1 loss are "HR-mediated UFBs." While logical, there are many problems with this conclusion.

1) The control cells also show predominantly FANCD2- UFBs, so it is unclear that 53BP1 has anything to do with this.

Our Response:

We are not sure which control cells the reviewer refers to in this case. However, if it is referred to the cells treated with siControl oligo, they are HeLa 53BP1^{hypo} cells, which as we found (partial) inactivation of 53BP1 increases the formation of FANCD2 –ve UFB (Figure 4i, also see new Figure 4a). Thus, this indicates that 53BP1 is required to suppress the excessive formation of FANCD2 –ve UFBs and that such an increase is dependent upon RAD51 (Fig. 4i).

If the control cells are referred to the parental HeLa (Fig. 4a), this suggests there are spontaneous formation of FANCD2 –ve UFBs even in the presence of 53BP1. They could be endogenous levels of catenation structures or HR/replication intermediates in these cancerous cell lines.

2) It is unclear what FANCD2 localization is reflecting in this experiment.

Our Response:

Based on our and other previous studies, FANCD2/I localises to the termini of UFB under replication stress conditions such as low doses of aphidicolin treatment. Therefore, it is used as a marker for (late) replication intermediate structures. Crucially, the formation of FANCD2⁺ve UFB is independent of RAD51, suggesting they are unlikely representing recombination intermediates. Indeed, losing RAD51 (HR pathway) increases FANCD2⁺ve UFBs, suggesting HR pathway prevents replication stress-associated structures. Based on these, we believe FANCD2 is a very useful marker to label replication intermediate-induced UFBs. Therefore, the absence of FANCD2 on UFBs in 53BP1-depleted cells may indicate that there is a distinct subclass of replication intermediates or HR structures. Given that the increased formation of FANCD2-negative UFBs in 53BP1^{hypo} cells is dependent on RAD51, this makes us to believe that HR pathway is involved at least to some extent as also suggested by the reviewer (below).

Certainly, one possibility is that 53BP1 suppresses UFBs that are mediated by RAD51. However, a challenge to this interpretation is that both 53BP1 loss and RAD51 disruption cause elevated UFBs, which makes it difficult to ascribe the order of events.

For example, another possibility is that 53BP1 be acting upstream to increase replication stress, and then RAD51 is important to convert such events into FANCD2⁻. Similarly, 53BP1 could be causing a different type of replication stress compared to Aphidicolin, where again 53BP1 could be acting upstream, and then APH (Note: RAD51?) could convert such events to FANCD2⁻ intermediates.

Our Response:

We agree with the reviewer that, in addition to the possibility that 53BP1 may negatively regulate HR activities, 53BP1 may also act upstream to prevent or solve replication problems that cause the formation of a type of replication intermediates (but not recognised by FANCD2) and subsequently converted into the FANCD2-negative sister DNA bridges in a RAD51-dependent manner. The replication intermediates, if exist, may also contribute to the increased formation of FANCD2-ve UFBs in 53BP1-deficient cells. We have now elaborated on these possibilities in our revised manuscript (pages 8 & 13; Fig. 4j & Supplementary Fig. 12). However, given

that RAD51 knockdown can specifically reduce the FANCD2-ve UFBs, it seems that HR pathway contributes to the accumulation of the sister DNA bridging structures to some extent as mentioned by the reviewer. We now include a possible model in new Supplementary Fig. 12 to depict these possibilities.

The contradiction is similar to that observed with loss of BLM and RAD51, where both cause an increase in UFBs, but only the former causes elevated sister chromatid exchanges (i.e. elevated SCEs don't necessarily mean more HR, but can reflect elevated crossing over).

Our Response:

We agree with the reviewer's comments. However, we think the situation is more complicated with BLM and RAD51 on UFB formation. Since BLM, but not 53BP1, also localises to UFBs and it has been suggested for UFB resolution, the elevation of UFBs in BS cells could be due to the lack of (1) UFB resolution, (2) lack of dHJ dissolution activity that may increase HJ accumulation and/or (3) increase the formation of replication intermediates. Nevertheless, we agree that the increases of SCEs in 53BP1 hypo cells could be due to lack of non-crossing over resolution activities or increased HR reaction. We have included these possibilities in the revised manuscript (page 8).

Plus, there is data in the literature supporting a role of 53BP1 in mediating appropriate HR, which contradicts their assertion of a simple HR-inhibition role of 53BP1.

Our Response:

We agree that 53BP1 may promote HR repair fidelity, maybe mainly on double-ended DSBs likely by preventing excessive single stranded annealing (SSA) reaction through blocking exacerbated DNA end resection. We have now included our opinion in the text (page 8) to state that 53BP1 may play differently in HR control on double-ended and single-ended DSBs, where the latter associates with damaged forks and is unlikely an ideal substrate for SSA. If excessive DNA end resection occurs at single-ended DSBs, it may not be sufficient to channel into SSA and suppress HR fidelity as there is no other recipient complementary ends for SSA. On the other hand, this may increase the opportunities of HR reaction and the increased formation of sister DNA intertwinements. Nevertheless, we agree that our proposed HR-inhibition role of 53BP1 is one of the possible causes of the elevation of sister chromatid bridges.

The bottom line here is that I agree with the prior reviewers that the authors have over-interpreted the mechanistic insight in their findings, which in fact may limit the overall impact of the work in the long run. I suggest that the authors focus on the interesting observation that 53BP1 loss causes a distinct subset of UFBs compared to APH treatment or RAD51 depletion, and then provide an even-handed discussion of possible mechanistic explanations, which includes reworking the Title, etc...

Thanks for the suggestion. We have now re-worded our text (revised discussion) and included other possible explanations of the formation of sister DNA bridges in 53BP1-depleted cells as per the suggestion. We also replaced/removed "HR-mediated UFB" to FANCD2-negative sister DNA bridges and amended the claim of "cancer cell specific" throughout the text.

Reviewers' Comments:

Reviewer #1:

Remarks to the Author:

In the revised manuscript by Tiwari and colleagues, entitled "53BP1 can limit sister-chromatid rupture and rearrangements driven by a distinct ultrafine DNA bridge-breakage pathway", the authors have addressed the concerns I had raised in the previous round of review. In particular they have amended the assertion that these bridges are a general feature of cancer cells and have included additional controls as requested. I would therefore recommend this manuscript for publication in its revised form.

Reviewer #2:

Remarks to the Author:

In their revised manuscript, Kok and co-workers have addressed my comments. I am satisfied with all the additional statistical analysis, and I am pleased to see that the wording throughout the manuscript has been changed.

An issue that I still find difficult, is that I don't think the mechanisms that underpin this phenomenon have been sufficiently worked out. However, I feel that in large part this is outside the scope of the current paper. Having said that, I do find that the wording at multiple points in the manuscript is still too strong:

For instance:

Page 12: 'delineated a distinct sister chromatid rupture and bridging pathway'

Page 15: 'identify a novel mitotic chromatid rupture-bridging pathway'

Since it remains unclear at what stage 53BP1 exactly operates, I do not think you can state that the pathway is identified/delineated. Also, the paper describes a phenotype that occurs in aberrant situations. In my opinion, that is not a 'pathway' but a 'consequence'/'phenotype', and the text should be changed on these aspects.

Reviewer #3:

Remarks to the Author:

My concerns have been adequately addressed, regarding the additional information provided in the new Fig 4A, as well as the additional discussion points, alternative models/conclusions, and reshaping/focusing several of the summary statements, including the title. I think the findings in this manuscript will contribute to the research on sources/mechanisms of anaphase chromosomal instability.

REVIEWERS' COMMENTS:

Reviewer #1 (Remarks to the Author):

In the revised manuscript by Tiwari and colleagues, entitled "53BP1 can limit sister-chromatid rupture and rearrangements driven by a distinct ultrafine DNA bridge-breakage pathway", the authors have addressed the concerns I had raised in the previous round of review. In particular they have amended the assertion that these bridges are a general feature of cancer cells and have included additional controls as requested. I would therefore recommend this manuscript for publication in its revised form.

We thank you for the reviewer's positive comments and recommendation.

Reviewer #2 (Remarks to the Author):

In their revised manuscript, Kok and co-workers have addressed my comments. I am satisfied with all the additional statistical analysis, and I am pleased to see that the wording throughout the manuscript has been changed.

An issue that I still find difficult, is that I don't think the mechanisms that underpin this phenomenon have been sufficiently worked out. However, I feel that in large part this is outside the scope of the current paper. Having said that, I do find that the wording at multiple points in the manuscript is still too strong:

For instance:

Page 12: 'delineated a distinct sister chromatid rupture and bridging pathway'

Page 15: 'identify a novel mitotic chromatid rupture-bridging pathway'

Since it remains unclear at what stage 53BP1 exactly operates, I do not think you can state that the pathway is identified/delineated. Also, the paper describes a phenotype that occurs in aberrant situations. In my opinion, that is not a 'pathway' but a 'consequence'/'phenotype', and the text should be changed on these aspects.

We thank you for the suggestion. In the revised manuscript we have replaced words such as 'delineate and identify' to 'reveal, describe and show'. Since our study does identify a new process of chromatid rupture/damage triggered by sister DNA bridging, which is distinct from the well-known breakage-fusion-bridge (BFB) mechanism, we believe that 'pathway' is a suitable term. However, we understand reviewer's concern, we now replace the term 'chromatid rupture/bridging pathway' into '...rupture/bridging phenomenon, process or action' throughout the manuscript and in the title as well.

Reviewer #3 (Remarks to the Author):

My concerns have been adequately addressed, regarding the additional information provided in the new Fig 4A, as well as the additional discussion points, alternative models/conclusions, and reshaping/focusing several of the summary statements, including the title. I think the findings in this manuscript will contribute to the research on sources/mechanisms of anaphase chromosomal instability.

We thank you for the reviewer's positive comments.